# TimeSearch-R: Adaptive Temporal Search for Long-Form Video Understanding via Self-Verification Reinforcement Learning

**Junwen Pan**[1*]   **Qizhe Zhang**[1,2*]   **Rui Zhang**[1]   **Ming Lu**[2]
**Xin Wan**[1]   **Yuan Zhang**[1,2]   **Chang Liu**[1]   **Qi She**[1†]
[1] ByteDance    [2] School of Computer Science, Peking University
{panjunwen,sheqi.roger}@bytedance.com

## Abstract

Temporal search aims to identify a minimal set of relevant frames from tens of thousands based on a given query, serving as a foundation for accurate long-form video understanding. Many existing works attempt to progressively narrow the search space. However, these approaches typically rely on a hand-crafted search process, lacking end-to-end optimization for learning optimal search strategies. In this paper, we propose **TimeSearch-R**, which reformulates temporal search as interleaved text–video thinking, seamlessly integrating searching video clips into the reasoning process through reinforcement learning (RL). However, applying RL training methods, such as Group Relative Policy Optimization (GRPO), to video reasoning can result in unsupervised intermediate search decisions. This leads to insufficient exploration of the video content and inconsistent logical reasoning. To address these issues, we introduce GRPO with Completeness Self-Verification (GRPO-CSV), which gathers searched video frames from the interleaved reasoning process and utilizes the same policy model to verify the adequacy of searched frames, thereby improving the completeness of video reasoning. Additionally, we construct datasets specifically designed for the SFT cold-start and RL training of GRPO-CSV, filtering out samples with weak temporal dependencies to enhance task difficulty and improve temporal search capabilities. Extensive experiments demonstrate that TimeSearch-R achieves substantial improvements on temporal search benchmarks such as Haystack-LVBench and Haystack-Ego4D, long-form video understanding benchmarks like VideoMME, MLVU, and LongVideoBench, as well as video reasoning benchmarks such as Video-Holmes, consistently and significantly outperforming other existing temporal search approaches and text-only reasoning models. *Our code is available at https://github.com/Time-Search/TimeSearch-R.*

## 1 Introduction

Long-form video understanding requires models to navigate through tens of thousands of frames to identify the most relevant information for answering specific questions (Fu et al., 2024; Zhou et al., 2024; Wu et al., 2024). Temporal search lies at the heart of making long-video understanding both accurate and interpretable (Park et al., 2025; Li et al., 2023; Ye et al., 2025). In contrast to the human visual system, which conducts adaptive temporal search (Yarbus, 1967; Hayhoe & Ballard, 2005), current large video-language models (LVLMs) primarily rely on hand-crafted search strategies with static frame sampling (Lin et al., 2023; Bai et al., 2025; Feng et al., 2025). Humans naturally alternate between broad scanning and targeted inspection, refining their focus iteratively based on intermediate findings (Castelhano & Henderson, 2007; Henderson & Hayes, 2017). In contrast, existing methods are limited to a fixed set of frames established before the reasoning process begins. This design presents a fundamental contradiction: video reasoning is a dynamic process where temporal search interleaves with video reasoning; however, the video frames accessible to the model remain fixed from the outset, ultimately hindering effective reasoning.

---

[*]Equal contribution.
[†]Corresponding author.

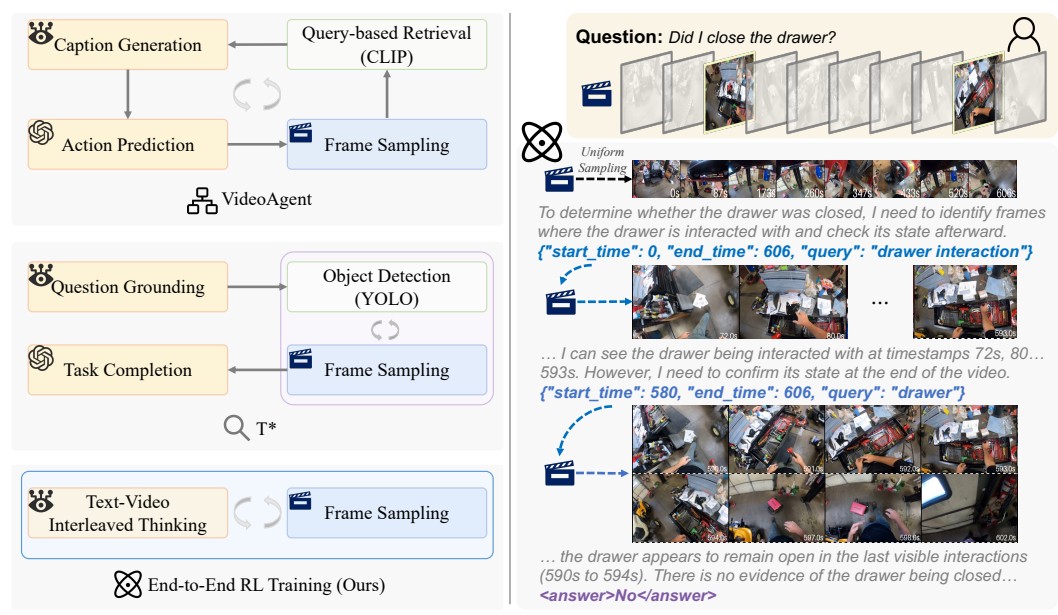

**(a) Paradigms of Temporal Search**  **(b) Text-Video Interleaved Thinking**

Figure 1: **(a) Different paradigms of temporal search.** Previous works such as VideoAgent (Wang et al., 2024) and T* (Ye et al., 2025) predominantly rely on handcrafted workflows, resulting in suboptimal strategies. Our approach adopts end-to-end reinforcement learning, enabling the model to learn optimal search strategies directly from data. **(b) Interleaved text–video thinking process.** We reformulate the temporal search task as an interleaved text–video thinking process, where the temporal search is seamlessly interleaved into the reasoning process.

Inspired by the gap between human cognition and model reasoning, recent studies have explored interactive video agents that attempt to bridge this divide through multi-turn temporal search, as illustrated in Figure 1 (a). VideoAgent (Wang et al., 2024) first employs a large language model (LLM) as the central agent, which iteratively calls tools like vision–language models (VLMs) and CLIP (Radford et al., 2021) for frame captioning and retrieval, and then aggregates information in the textual modality to perform reasoning and predict answers. T* (Ye et al., 2025) extends this paradigm by introducing an object-oriented spatial-temporal search. It first leverages a VLM to extract target objects from the question, then employs object detection models (e.g., YOLO (Cheng et al., 2024)) to identify keyframes containing these objects, and finally uses the retrieved frame set to complete the task. Moreover, strategies that introduce tree-structured search to improve efficiency have also been explored (Wang et al., 2025; Li et al., 2025; Pan et al., 2025). However, all of these approaches depend on manually designed workflows, which lead to suboptimal search strategies.

This motivates us to explore an end-to-end learning approach that discovers optimal temporal search strategies directly from data. In this work, we reformulate the temporal search task as an interleaved text–video thinking process, and propose **TIMESEARCH-R**, a model that learns to actively search for relevant temporal clips through reinforcement learning (RL). As shown in Figure 1 (b), our model alternates between textual reasoning and temporal exploration, iteratively refining its understanding of the video. We refer to this dynamic process as **Thinking with Videos**—a paradigm where models gradually improves their comprehension by searching for relevant video content conditioned on intermediate reasoning states. This concept extends the recent advances in multimodal reasoning, *Thinking with Images* (Hu et al., 2024; Zheng et al., 2025), to the long-video domain.

Although recent works have successfully applied RL algorithms like Group Relative Policy Optimization (GRPO) (DeepSeek-AI, 2025) to textual (Jin et al., 2025) and spatial search (Zheng et al., 2025), temporal search in videos poses unique challenges. The original GRPO rewards only the final output while ignoring intermediate search decisions, leading to several failure modes illustrated in Figure 2. The first mode, termed **insufficient temporal exploration**, arises because the final output reward provides no incentive for comprehensive exploration of video frames. LVLMs may arrive at correct answers through partial evidence or language bias without proper visual grounding (Niu et al., 2021), missing critical frames required for reliable understanding. The second mode, termed **inconsistent**

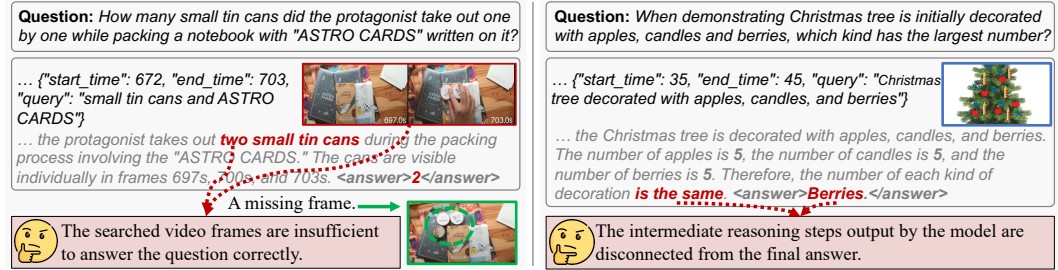

Figure 2: **Two failure modes with the original GRPO reward.** Left: **Insufficient temporal exploration.** The model misses critical frames required to correctly answer the question. Right: **Inconsistent logical reasoning.** The intermediate reasoning process contradicts the final answer.

**logical reasoning**, emerges when models produce plausible thinking processes disconnected from the final answers, a phenomenon also observed in text-only reasoning (Lanham et al., 2023). These two failure modes hinder proper temporal search and diminish the benefits of video reasoning.

To address these challenges, we propose **Completeness Self-Verification (CSV)** as a supplement to the original GRPO algorithm, providing supervision over the intermediate steps of temporal search. GRPO-CSV tackles insufficient temporal exploration by ensuring the model to acquire sufficient visual evidence through self-verification, and promotes consistency between intermediate reasoning and the final answer by re-answering the question using the searched frames. Besides, we construct a high-quality video reasoning dataset to support GRPO-CSV training. Existing datasets contain a large number of trivial samples solvable through pure linguistic bias, as well as noisy samples that remain unsolvable even with extensive search, severely hindering progress in long-video reasoning. We implement a two-stage data filtering pipeline to curate high-quality samples tailored to the demands of video reasoning, ensuring that the model learns the correct process of temporal search.

We evaluate our TimeSearch-R on both temporal search and long-form video understanding tasks, demonstrating its superiority in long video reasoning. On temporal search tasks, TimeSearch-R improving the temporal F1 score on Haystack-LVBench by 5.9% and the accuracy on Haystack-Ego4D by 8.3%, compared to the previous state-of-the-art method. On long-form video understanding tasks, TimeSearch-R consistently surpasses existing temporal search methods and the advanced reasoning model Video-R1. Moreover, the learned search policy transfers readily to other foundation models like GPT-4o, achieving significant performance improvements at a lower cost. On video reasoning tasks, TimeSearch-R achieves an 11.8% improvement over the base model Qwen2.5-VL on Video-Holmes, and even outperforms advanced proprietary models such as Gemini-2.0.

In summary, our main contributions are three-fold:

1. We propose the **TimeSearch-R** framework, which reformulates temporal search as interleaved text–video thinking and learns optimal search strategies directly from data.

2. We introduce **GRPO-CSV**, a novel RL algorithm, which ensures sufficient and accurate video exploration by supervising the intermediate steps of temporal search. To support GRPO-CSV training, we also construct a high-quality video reasoning dataset via a two-stage filtering pipeline, enabling the model to learn correct temporal search processes.

3. Extensive experiments demonstrate the superiority of our approach on both temporal search and long-form video understanding. TimeSearch-R consistently and significantly outperforms other existing temporal search methods and advanced reasoning models.

## 2 METHODS

In this section, we first reformulate the temporal search task as an interleaved text–video thinking process, enabling the model to learn optimal search strategies directly from data. To address the challenges of insufficient temporal exploration and inconsistent logical reasoning, we introduce GRPO-CSV as a novel RL algorithm for long videos, which ensures both sufficient and accurate video exploration by supervising intermediate steps of temporal search. Finally, we describe the model training process, including the construction of a high-quality long-video reasoning dataset.

## 2.1 TASK FORMULATION

**Temporal Search within Thinking Process.**   To learn optimal search strategies directly from data, we reformulate temporal search as a multi-turn thinking process interleaved with video clip retrieval. Given a video $V$ and a corresponding question $Q$, an initial preview $\tilde{V}$ is uniformly sampled from $V$ for subsequent reasoning. At each thinking step $k$, the policy model $\pi_\theta$ generates a textual reasoning $T_k$. If $T_k$ contains a `search` instruction, the video environment executes it according to frame timestamps, retrieving a clip $V_k \subseteq V$ that is appended to the ongoing chain of thought (CoT) as input for later steps. The interleaved text-video CoT at reasoning step $k$ is formalized as:

$$C_k \triangleq \{ (T_1, V_1), (T_2, V_2), \ldots, (T_k, V_k) \}. \tag{1}$$

This interaction process repeats until the model emits the final answer $A$ or reaches the pre-defined reasoning budget. For further analysis, the entire reasoning chain can be decomposed into two components: temporal search and answer prediction, which can be formulated as:

$$P_\theta(A, C \mid \tilde{V}, Q) = \underbrace{P_\theta(C \mid \tilde{V}, Q)}_{\text{Temporal Search}} \cdot \underbrace{P_\theta(A \mid C, \tilde{V}, Q)}_{\text{Answer Prediction}}. \tag{2}$$

**Dynamic Video Frames.**   During the interleaved thinking process, the model autonomously explores the video by searching for additional clips. At reasoning step $k$, if the model outputs a `search` instruction, it is also required to specify the temporal boundaries $t_s^k$ and $t_e^k$ to be explored, along with a corresponding textual query $q^k$. The video environment then executes a frame retrieval function to obtain additional $F$ frames $V_k = \texttt{search}(V; t_s^k, t_e^k, q^k, F) = \{f_k^1, f_k^2, \ldots, f_k^F\}$. This function serves as an interface to the policy model $\pi_\theta$, employing a small VLM (e.g., SigLIP (Zhai et al., 2023)) to calculate the similarity among frames within the specified temporal clip $[t_s^k, t_e^k]$, as well as the relevance with the textual query $q^k$. The most informative $F$ frames are then sampled using determinantal point process (DPP) (Kulesza & Taskar, 2012), which has been widely used for information retrieval (Chen et al., 2018; Celis et al., 2018; Sun et al., 2025). This operation significantly improves the efficiency of temporal search, and more details can be found in Section A.

## 2.2 GRPO WITH COMPLETENESS SELF-VERIFICATION

Evaluating temporal search typically requires frame-level annotations (Ye et al., 2025), which are time-consuming and labor-intensive. Previous works (Yu et al., 2025b; Sun et al., 2025) evaluate the selected frame sets in downstream video understanding tasks as a surrogate metric. Inspired by this, we design a **Completeness Self-Verification (CSV)** mechanism for GRPO, which is annotation-free and can be seamlessly integrated into RL training, serving as a complementary to the original outcome reward. The overall pipeline of GRPO-CSV is illustrated in Figure 3.

**GRPO-CSV.**   We introduce CSV as a complement to GRPO with only outcome rewards. During the GRPO rollout phase, the policy model $\pi_\theta$ generates a text–video interleaved CoT $C$ and a final answer $A$. Applying rewards only to the final answer may reduce the effectiveness of intermediate search processes. To address this, we extract the video clips from $C$ to form a dynamic frame set $V_c$ as the input for the CSV phase. In the CSV rollout phase, the same policy model is required to re-answer the question $Q$ using only $V_c$, yielding a CSV answer $A_c$. Critically, the model is prohibited from any further temporal searching and must rely solely on the currently searched frames to answer the question. The CSV answer $A_c$ is expected to remain consistent with the original answer $A$:

$$P_\theta(A_c \mid V_c, Q) \approx P_\theta(A \mid C, \tilde{V}, Q). \tag{3}$$

**Completeness Reward.**   We design a completeness reward for the CSV phase, which is computed using the original answer $A$, the CSV answer $A_c$, and the ground-truth answer $A^*$ as follows:

$$R_c = \mathbb{1}[\text{Acc}(A, A^*) > 0.5] \cdot \text{Acc}(A_c, A^*). \tag{4}$$

where $\text{Acc}(A, A^*)$ and $\text{Acc}(A_c, A^*)$ denote the correctness scores of the original answer and the CSV answer, respectively, and $\mathbb{1}[\cdot]$ is an indicator function activated only when the original answer $A$ is correct. This conditional design ensures that the CSV reward is applied only to promising reasoning trajectories, encouraging meaningful temporal search while verifying both the sufficiency of acquired visual evidence and the consistency between the reasoning process and the final answer.

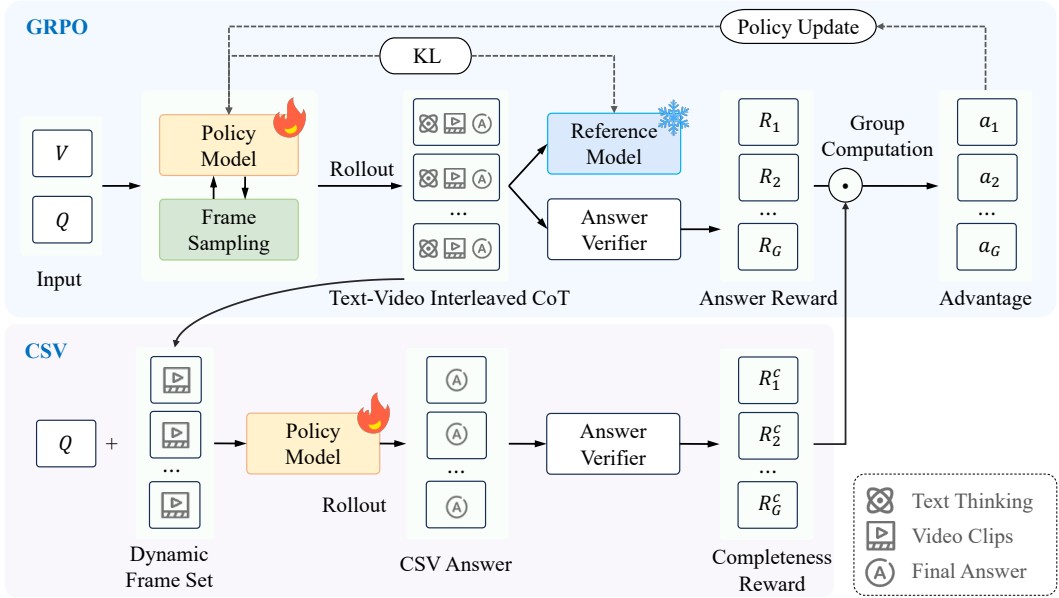

Figure 3: **Overall pipeline of GRPO-CSV.** Building upon the original GRPO, CSV extracts a dynamic frame set from the multi-modal CoT and constructs a vision-only CoT for re-answering. This design verifies that the searched dynamic frames provide sufficient evidence for correct reasoning, ensuring completeness and consistency without requiring explicit frame-level supervision.

From an information-theoretic perspective (Shannon, 1948), relying solely on the final answer reward primarily maximizes the mutual information $I(A; Q)$, making the final answers strongly dependent on the questions, while neglecting $I(A; V)$, the dependence of answers on the video input. This permits the model to exploit language biases as shortcuts instead of performing sufficient video exploration. Our completeness reward enforces high $I(A; V_c)$, which reinforces the pathway from $V$ to $A$, thereby increasing $I(A; V)$ and ensuring that the model performs effective temporal search.

From an optimization perspective, using only the final answer reward leads to an extremely sparse binary signal and a vanishing-advantage problem (Yu et al., 2025a). Moreover, a correct final answer may arise from lucky guessing, yet receives the same incentive as one derived from proper temporal search, making it difficult to learn how to utilize video content. The completeness reward supplements correct answers with supervision over the intermediate reasoning process, transforming the originally sparse reward into a denser signal, enabling better credit assignment and model optimization.

**Format Reward.** The format reward enforces adherence to a predefined schema throughout the multi-turn reasoning process, validating the structural integrity of the entire trajectory rather than individual steps. During reasoning, each step must follow either the `<think>...</think><tool_call>...</tool_call>` pattern for temporal search or the `<think>...</think><answer>...</answer>` pattern for the final response. We assign a binary score to the full trajectory: 1 if all steps are structurally valid, and 0 otherwise.

**Accuracy Reward.** We evaluate answer accuracy for two task types. For multiple-choice questions, we extract the option letter from the model's output and perform an exact match with the ground-truth option. For open-ended questions, we adopt an LLM-as-a-Judge approach (Zheng et al., 2023) to assess the semantic agreement between the model's final answer and the reference answer. The scores for both cases are given in binary form, with 1 indicating alignment with the standard answer.

**Overall Reward.** The total reward is the sum of completeness, format, and accuracy components:

$$R = R_c + R_{\text{fmt}} + R_{\text{acc}}. \tag{5}$$

This composition encourages sufficient temporal exploration ($R_c$), consistent reasoning structures ($R_{\text{fmt}}$), and correct final answers ($R_{\text{acc}}$), enhancing the model's ability to understand long-form videos.

Table 1: **Temporal search performance.** We report temporal similarity, visual similarity, and question-answering (QA) accuracy on Haystack-LVBench, as well as QA accuracy on Haystack-Ego4D test-tiny subset. Baseline results are directly cited from Ye et al. (2025). [†] indicates the average number of keyframes determined by the model adaptively.

| Method | Base Model | # Keyframes | Temporal | | | Visual | | | QA | |
|---|---|---|---|---|---|---|---|---|---|---|
| | | | P | R | $F_1$ | P | R | $F_1$ | LVBench | Ego4D |
| *Static Frame Sampling* | | | | | | | | | | |
| Uniform | GPT-4o | 8 | 1.4 | 6.3 | 2.2 | 56.0 | 72.0 | 62.7 | 47.1 | 41.5 |
| Uniform | GPT-4o | 32 | 1.4 | 24.9 | 2.7 | 58.7 | 81.6 | 67.3 | 50.5 | 45.5 |
| *Adaptive Temporal Search* | | | | | | | | | | |
| VideoAgent (Wang et al., 2024) | GPT-4 | 10.1[†] | 1.2 | 8.5 | 2.1 | 58.8 | 73.2 | 64.7 | – | – |
| Retrieval-based (Ye et al., 2025) | GPT-4o | 8 | 1.5 | 6.3 | 2.3 | 63.1 | 65.5 | 64.1 | – | – |
| T* (Ye et al., 2025) | GPT-4o | 8 | 1.6 | 7.1 | 2.5 | 58.4 | 72.7 | 64.3 | 51.9 | 45.0 |
| Retrieval-based (Ye et al., 2025) | GPT-4o | 32 | 1.3 | 21.8 | 2.4 | 59.9 | 80.8 | 67.8 | – | – |
| T* (Ye et al., 2025) | GPT-4o | 32 | 1.7 | 28.2 | 3.1 | 58.3 | 83.2 | 67.8 | 53.1 | 46.5 |
| *Text-Video Interleaved Reasoning* | | | | | | | | | | |
| TimeSearch-R (Ours) | Qwen2.5-VL-7B | 7.8[†] | 5.8 | 21.5 | 8.4 | 63.2 | 75.8 | 69.0 | 52.4 | 53.3 |
| TimeSearch-R (Ours) | Qwen2.5-VL-7B | 30.7[†] | 4.2 | 36.8 | 7.0 | 63.1 | 81.6 | 71.2 | – | – |

## 2.3 MODEL TRAINING

**Dataset Construction.** A fundamental challenge in RL for long-video reasoning lies in the fact that a large number of samples in the existing datasets can be solved through pure linguistic bias, reducing the reliance on temporal search. Moreover, some noisy samples remain unsolvable even under ideal temporal search, preventing the model from effectively exploring the video. To address these challenges, we implement a two-stage data filtering pipeline to construct a high-quality dataset tailored to video reasoning. In the first stage, we remove samples that the policy model can solve correctly using only 4 uniformly sampled frames, thereby discouraging reliance on linguistic shortcuts. In the second stage, we further discard samples that remain unsolvable even with multiple temporal searches and numerous video frames, ensuring active video exploration. Additional details of this filtering pipeline are provided in Section B.1. And we enhance dataset diversity through the incorporation of samples sourced from Haystack-Ego4D (Ye et al., 2025), VideoMarathon (Lin et al., 2025), and CinePile (Rawal et al., 2024). A detailed analysis of the dataset is presented in Section B.2.

**Model Training.** We employ a two-stage training scheme for our TimeSearch-R. In the first stage, supervised fine-tuning (SFT) serves as a cold start, guiding the model to follow the correct reasoning format and enabling effective policy learning in the subsequent RL stage. SFT training adopts the above dataset construction pipeline, using GPT-4o (OpenAI, 2024) to generate the text-video interleaved reasoning processes and the corresponding final answers. Following practices in the text domain (Jin et al., 2025), we mask the temporal search results during training to force the model to learn meaningful temporal windows and textual queries. The objective in this stage is to minimize the standard cross-entropy loss over reasoning tokens, while excluding masked video tokens from gradient computation. Building on this cold-start, we further conduct RL post-training based on the proposed GRPO-CSV algorithm to stimulate the temporal reasoning capability of the model.

## 3 EXPERIMENTS

### 3.1 EXPERIMENTAL SETUP

**Baselines.** To comprehensively evaluate the effectiveness of TimeSearch-R, we compare it against three types of baselines: (1) Advanced foundation models with static frame sampling, including both API models (OpenAI, 2024; Team et al., 2024) and open-source models (Bai et al., 2025). (2) State-of-the-art temporal search agents, such as VideoAgent (Wang et al., 2024), T* (Ye et al., 2025), and VideoTree (Wang et al., 2025). (3) Video reasoning models like Video-R1 (Feng et al., 2025).

**Datasets.** We evaluate TimeSearch-R on three tasks: (1) Temporal search on Haystack-LVBench and Haystack-Ego4D (Ye et al., 2025), where the task is modeled as long video needle-in-a-haystack, measuring temporal and visual similarity as well as QA accuracy. (2) Long-form video understanding on VideoMME (Fu et al., 2024), MLVU (Zhou et al., 2024), and LongVideoBench (Wu et al., 2024). (3) Complex video reasoning on Video-Holmes (Cheng et al., 2025).

Table 2: **Video understanding performance.** E2E stands for end-to-end optimization. $^\dagger$ indicates the average number of frames determined by the model adaptively. *Temporal Search Transfering* refers to transferring the learned temporal search strategies (*i.e.* searched dynamic frames $V_c$ in Equation 3) from TimeSearch-R to other foundation models (*e.g.* GPT-4o).

| Model | E2E | # Frame | VideoMME (w/o sub) | | | | MLVU | LVB |
| | | | short | medium | long | overall | m-avg | val |
|---|---|---|---|---|---|---|---|---|
| *Static Frame Sampling* | | | | | | | | |
| Qwen2.5-VL-7B (Bai et al., 2025) | ✓ | 8 | 62.8 | 54.0 | 47.3 | 54.7 | 54.3 | 54.2 |
| GPT-4o (OpenAI, 2024) | ✓ | 8 | 72.2 | 62.7 | 56.8 | 63.9 | 58.1 | 56.7 |
| LLaVA-OV-72B (Li et al., 2024) | ✓ | 8 | 68.0 | 58.2 | 55.2 | 60.5 | 62.8 | 56.8 |
| Qwen2.5-VL-7B (Bai et al., 2025) | ✓ | 32 | 73.5 | 58.6 | 51.7 | 61.2 | 58.1 | 58.2 |
| GPT-4o (OpenAI, 2024) | ✓ | 32 | 79.8 | 69.7 | 62.7 | 70.8 | 66.6 | 61.1 |
| LLaVA-OV-72B (Li et al., 2024) | ✓ | 32 | 77.2 | 63.6 | 58.4 | 66.4 | 69.2 | 62.4 |
| *Text-only Reasoning* | | | | | | | | |
| Video-R1-7B (Feng et al., 2025) | ✓ | 16 | 67.2 | 58.0 | 48.9 | 58.0 | 60.9 | 53.0 |
| Video-R1-7B (Feng et al., 2025) | ✓ | 32 | 71.1 | 59.0 | 49.4 | 59.9 | 61.6 | 56.4 |
| *Adaptive Temporal Search* | | | | | | | | |
| VideoAgent (Wang et al., 2024) | ✗ | 87$^\dagger$ | – | – | 49.0 | 56.0 | – | – |
| VideoTree (Wang et al., 2025) | ✗ | 128 | 67.8 | 59.9 | 54.2 | – | – | – |
| T* + GPT-4o | ✗ | 8 | 56.4 | 57.3 | 55.9 | 56.5 | – | – |
| T* + LLaVA-OV-72B | ✗ | 8 | 61.7 | 57.5 | 57.7 | 59.0 | – | – |
| T* + GPT-4o | ✗ | 32 | 69.5 | 63.5 | 59.3 | 64.1 | – | – |
| T* + LLaVA-OV-72B | ✗ | 32 | 77.5 | 66.6 | 61.0 | 68.3 | – | – |
| *Text-Video Interleaved Reasoning* | | | | | | | | |
| TimeSearch-R-7B (Ours) | ✓ | 7.8$^\dagger$ | 71.0 | 56.8 | 49.6 | 59.1 | 68.6 | 57.1 |
| TimeSearch-R-7B (Ours) | ✓ | 15.9$^\dagger$ | 74.1 | 61.0 | 53.0 | 62.7 | 70.7 | 58.6 |
| TimeSearch-R-7B (Ours) | ✓ | 31.9$^\dagger$ | 76.0 | 63.0 | 53.3 | 64.1 | 71.3 | 61.6 |
| *Temporal Search Transfering* | | | | | | | | |
| TimeSearch-R-7B + GPT-4o | ✗ | 7.8$^\dagger$ | 76.7 | 68.6 | 60.1 | 68.5 | 68.6 | 60.8 |
| TimeSearch-R-7B + LLaVA-OV-72B | ✗ | 7.8$^\dagger$ | 72.3 | 62.6 | 57.8 | 64.2 | 71.5 | 61.1 |
| TimeSearch-R-7B + GPT-4o | ✗ | 31.9$^\dagger$ | 81.4 | 72.8 | 65.1 | 73.1 | 69.2 | 63.4 |
| TimeSearch-R-7B + LLaVA-OV-72B | ✗ | 31.9$^\dagger$ | 78.1 | 66.6 | 63.9 | 69.5 | 72.7 | 64.5 |

**Evaluation Metrics.** Besides the original metrics used in the benchmarks, we additionally introduce two metrics to assess the quality of the text–video interleaved thinking process for ablation study. Among them, *completeness* measures whether the searched frame set is sufficient for the correct answer, while *consistency* measures the alignment between intermediate reasoning and the final answer. Further details on these two metrics are provided in Section D.

**Implementation Details.** We train TimeSearch-R based on Qwen2.5-VL-7B-Instruct (Bai et al., 2025). In the RL training, we use the AdamW (Loshchilov & Hutter, 2017) optimizer with a learning rate of 1e-6, a KL penalty coefficient $\beta = 0.005$, and a batch size of 4 with 8 rollouts per prompt. We limit each search operation to retrieving at most 8 frames from a specified temporal clip, with up to 8 search steps in total. Training is conducted on 32 A100 GPUs. See more details in Section G. During inference, we control the target number of frames through the system prompt (Figure 8), and report the actual average frames used across test samples. See more details in Section C.

## 3.2 MAIN RESULTS

**Temporal Search.** On the temporal search task, TimeSearch-R establishes a new state-of-the-art on LV-Haystack, as shown in Table 1. Under a budget of 8 keyframes, our method achieves an $F_1$ score of 8.4 in temporal similarity, more than three times the previous best result of 2.5 obtained by T*. In visual similarity, TimeSearch-R reaches an $F_1$ score of 69.0, surpassing the previous SOTA method VideoAgent by 5.3, and even outperforming the retrieval-based method and T* with larger keyframe budgets. For the needle-in-a-haystack QA, our TimeSearch-R consistently outperforms the advanced API model GPT-4o, achieving 52.4% accuracy on Haystack-LVBench and 53.3% on Haystack-Ego4D, while maintaining a lower inference cost. These results demonstrate the superiority of end-to-end learned temporal search strategies over handcrafted workflows based on human heuristics.

Table 3: **Video reasoning performance on Video-Holmes.** SR: Social Reasoning; IMC: Intention & Motive Chaining; TCI: Temporal Causal Inference; TA: Timeline Analysis; MHR: Multimodal Hint Reasoning; PAR: Physical Anomaly Reasoning; CTI: Core Theme Inference.

| Model | #Frame | SR | IMC | TCI | TA | MHR | PAR | CTI | Overall |
|---|---|---|---|---|---|---|---|---|---|
| GPT-4o (OpenAI, 2024) | 32 | 50.0 | 49.6 | 38.8 | 30.0 | **44.0** | 39.2 | 37.0 | 42.0 |
| Gemini-2.0-Flash-Thinking | – | 43.4 | 46.9 | **43.1** | **51.0** | 37.9 | 43.6 | 39.3 | 43.1 |
| Qwen2.5-VL-7B (Bai et al., 2025) | 32 | 38.4 | 34.8 | 17.6 | 30.0 | 27.1 | 18.6 | 25.2 | 27.8 |
| Qwen2.5-VL-7B (Bai et al., 2025) | 768 | 34.1 | 42.6 | 29.9 | 18.5 | 30.7 | 32.5 | 33.0 | 32.1 |
| Video-R1-7B (Feng et al., 2025) | 32 | 48.6 | 41.7 | 28.9 | 34.5 | 31.0 | 33.5 | 35.9 | 36.5 |
| TimeSearch-R-7B (Ours) | 32 | **54.2** | 48.3 | 35.7 | 39.5 | 34.0 | 43.3 | 41.1 | 42.2 |
| TimeSearch-R-7B (Ours) | 768 | 53.9 | **52.7** | 33.2 | 40.0 | 40.1 | **44.9** | **42.2** | **43.9** |

| Method | Haystack-LVBench | | | VideoMME | | | |
|---|---|---|---|---|---|---|---|
| | P | R | $F_1$ | Comp. | Cons. | Acc. | |
| Qwen2.5-VL w/ search | 0.0 | 0.0 | 0.0 | 44.2 | 59.4 | 51.8 | |
| SFT | 7.4 | 11.6 | 7.8 | 60.5 | 69.2 | 59.2 | |
| GRPO (Before Collapse) | $5.2_{-2.2}$ | $18.8_{+7.2}$ | $7.4_{-0.4}$ | $57.2_{-3.3}$ | $69.3_{+0.1}$ | $65.1_{+5.9}$ | |
| GRPO-CSV w/o Acc. Rwd | $6.1_{-1.3}$ | $19.8_{+8.2}$ | $8.2_{+0.4}$ | $61.2_{+0.7}$ | $75.3_{+6.1}$ | $64.8_{+5.6}$ | |
| GRPO-CSV w/ Acc. Rwd | $5.4_{-2.0}$ | $22.3_{+10.7}$ | $8.1_{+0.3}$ | $60.2_{-0.3}$ | $71.8_{+2.6}$ | $66.6_{+7.4}$ | |

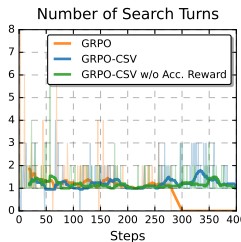

(a) Ablation Results        (b) Training Dynamics

Figure 4: **Ablation study of GRPO-CSV.** (a) Comparison of different training schemes on temporal search and long-form video understanding. (b) When CSV is removed, training begins to collapse. The model gradually reduces the number of search calls and eventually stops searching altogether.

**Long-Form Video Understanding.** Our TimeSearch-R also achieves strong performance on the long-form video understanding task, which is shown in Table 2. Under a budget of 8 input frames, our method outperforms the base model Qwen2.5-VL on VideoMME by 4.4%, and also surpasses the temporal search approach T* with the proprietary API model GPT-4o. When the budget increases to 32 frames, TimeSearch-R achieves substantial gains, exceeding the base model by 13.2% on MLVU and 3.4% on LongVideoBench. It also significantly outperforms VideoAgent and VideoTree, both of which use far more input frames, highlighting the advantages of end-to-end optimization. Compared with the latest video reasoning model Video-R1, our method consistently delivers superior performance across all three benchmarks under different frame budgets, validating that text–video interleaved reasoning is more effective than text-only reasoning for long-form video understanding. Notably, the search policy learned by TimeSearch-R transfers easily to other foundation models and yields substantial performance improvements, demonstrating strong generalization of the strategy.

**Complex Video Reasoning.** To evaluate the adaptability of TimeSearch-R, we conduct experiments on the video reasoning benchmark Video-Holmes, with results summarized in Table 3. Our method consistently outperforms the base model Qwen2.5-VL, the video reasoning model Video-R1, and the proprietary API models GPT-4o and Gemini-2.0, demonstrating its strong generality across tasks.

### 3.3 Ablation Studies

**Training Scheme.** We explore the impact of different training stages in Table 4a, from zero-shot CoT to SFT and finally RL, yielding two key findings: (1) **SFT enables search capability:** The model cannot perform the search well only through zero-shot CoT prompts. SFT allows the model to rapidly acquire temporal search skills, dramatically improving temporal $F_1$ from 0.0 to 7.8 and searched frame completeness from 44.2% to 60.5%. (2) **RL enhances video reasoning:** While RL provides modest improvements to temporal similarity and search completeness, its primary advantage lies in boosting overall understanding performance. The post-training stage improves reasoning consistency by 2.6%, which in turn raises QA accuracy from 59.2% to 66.6%.

**GRPO-CSV Component.** We further conduct an ablation study on the components of GRPO-CSV in Figure 4, and obtain three key findings: (1) **GRPO reduces search completeness.** Without CSV

Table 4: **Ablation study of data composition.** Line 1 shows the accuracy of original Qwen2.5-VL.

| Ego | Exo | Filter | General | | | | Reasoning | | | |
|---|---|---|---|---|---|---|---|---|---|---|
| | | | short | medium | long | overall | temporal | spatial | action | object |
| – | – | – | 76.3 | 66.0 | 54.6 | 65.1 | 51.4 | **76.8** | 56.8 | 59.5 |
| ✓ | ✓ | | 74.2 | 62.7 | 51.3 | 62.8 | 40.1 | 67.9 | 60.0 | 56.2 |
| ✓ | | ✓ | 76.4 | 64.7 | 54.9 | 65.3 | 54.8 | 73.2 | 58.2 | 59.0 |
| ✓ | ✓ | ✓ | **76.8** | **67.1** | **56.0** | **66.6** | **58.8** | 75.0 | **62.5** | **61.9** |

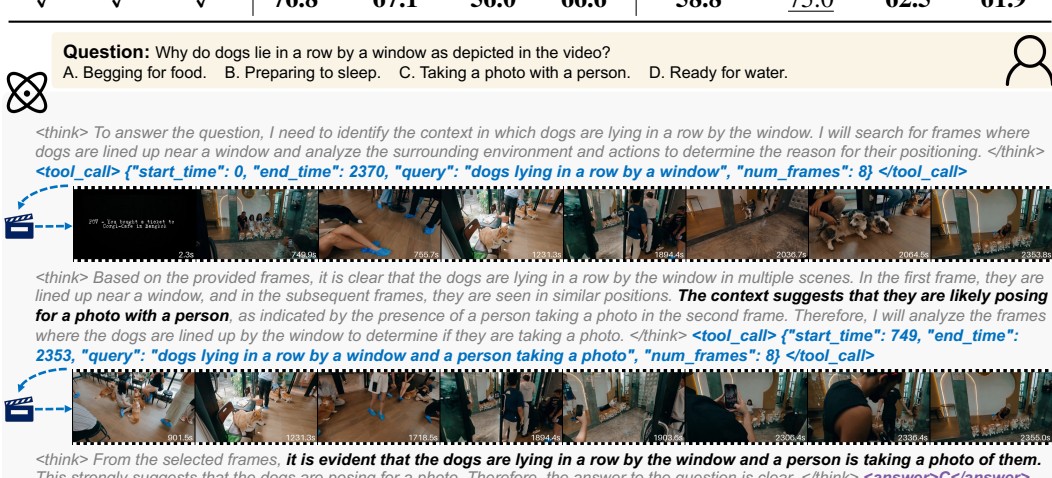

**Question:** Why do dogs lie in a row by a window as depicted in the video?
A. Begging for food.    B. Preparing to sleep.    C. Taking a photo with a person.    D. Ready for water.

*<think> To answer the question, I need to identify the context in which dogs are lying in a row by the window. I will search for frames where dogs are lined up near a window and analyze the surrounding environment and actions to determine the reason for their positioning. </think>*
*<tool_call> {"start_time": 0, "end_time": 2370, "query": "dogs lying in a row by a window", "num_frames": 8} </tool_call>*

*<think> Based on the provided frames, it is clear that the dogs are lying in a row by the window in multiple scenes. In the first frame, they are lined up near a window, and in the subsequent frames, they are seen in similar positions. **The context suggests that they are likely posing for a photo with a person**, as indicated by the presence of a person taking a photo in the second frame. Therefore, I will analyze the frames where the dogs are lined up by the window to determine if they are taking a photo. </think>* *<tool_call> {"start_time": 749, "end_time": 2353, "query": "dogs lying in a row by a window and a person taking a photo", "num_frames": 8} </tool_call>*

*<think> From the selected frames, **it is evident that the dogs are lying in a row by the window and a person is taking a photo of them.** This strongly suggests that the dogs are posing for a photo. Therefore, the answer to the question is clear. </think>* *<answer>C</answer>*

Figure 5: **Hypothesis-driven search.** Given the context that dogs are lying in a row across multiple scenes and remain still, the model hypothesizes that they are waiting to be photographed. It then searches for the person taking a photo to gather supporting evidence and provides the final answer.

as a complement, GRPO drops completeness from 60.5% to 57.2% and temporal $F_1$ from 7.8 to 7.4, demonstrating that outcome-only rewards lead to insufficient temporal exploration. (2) **GRPO-CSV improves training stability.** As illustrated in Figure4b, removing CSV causes training to collapse around step 300, after which the model ceases to make search calls and completeness drops to zero. (3) **GRPO-CSV with accuracy reward achieves the best QA performance.** While completeness reward alone achieves the highest completeness and consistency, it slightly reduces QA accuracy by 0.3%. Combining GRPO-CSV with accuracy reward leads to the best overall QA performance.

**Data Composition.** We also analyze the data composition in RL training, as shown in Table 4, revealing the contributions of data filtering and domain diversity. Without data filtering, RL training leads to a substantial performance drop compared to the original Qwen2.5-VL. This degradation arises because linguistic biases induce zero advantage in GRPO group computation: when questions can be trivially answered through linguistic shortcuts, all rollouts achieve perfect accuracy and completeness, yielding no learning signal and severely hindering RL efficiency and training stability. After applying data filtering, the model trained solely on egocentric data recovers baseline performance, but the lack of diversity weakens the benefits of RL. By incorporating exocentric data to enhance domain diversity, the model achieves its best general QA accuracy of 66.6%. Notably, although the training data only includes general long-video QA tasks, RL training significantly boosts the model's temporal and action reasoning capabilities, improving them by 7.4% and 5.7%, respectively. This remarkable performance demonstrates that TimeSearch-R learns fundamental cognitive patterns through end-to-end policy optimization, validating the strong generalization of our proposed GRPO-CSV algorithm.

## 3.4 CASE STUDIES

**Emergent Search Patterns.** We analyze the search patterns that emerge during end-to-end RL training, demonstrating how the model executes temporal search within its reasoning process in a manner analogous to human cognition. These search patterns exhibit adaptability and flexibility across different task types: **(1) Hypothesis-driven search:** The model formulates hypotheses based on limited context and executes targeted searches to gather additional frames as supporting evidence

(Figure 5). **(2) Confirmation or elimination:** When the initial frame set provides insufficient support for an answer, the model employs multi-faceted search strategies or elimination methods to collect additional evidence and reduce uncertainties (Figure 16 and 17). **(3) Sequential search:** The model performs segment-by-segment analysis to accomplish temporal reasoning tasks that require understanding sequential relationship in the video (Figure 18).

**Failure Cases.** The analysis of failure cases reveals several limitations that suggest directions for future work. **(1) Reasoning verbosity:** A small fraction of cases exhibit verbose or repetitive reasoning patterns (Figure 19), leading to search budget exhaustion before answering. **(2) Temporal reasoning errors:** Despite the strong temporal understanding capabilities of the base model, we observe occasional mistakes in time-related predictions, such as predicting timestamps that exceed the video duration (Figure 20). **(3) Reasoning inconsistency:** While our CSV module substantially improves reasoning completeness and consistency, we still observe instances of insufficient or inconsistent reasoning (Figure 21, 22 and 23). Evaluating video-text interleaved trajectories remains challenging because obtaining human-annotated ground truth is prohibitively expensive. **(4) Frame number prediction:** Determining the optimal number of frames appears overly difficult for current 7B-scale models, consequently retrieving irrelevant frames in some cases (Figure 24). These limitations highlight promising avenues for future research, including robust time prediction, efficient reasoning mechanisms, automatic evaluation of reasoning quality, and adaptive frame number prediction.

## 4 RELATED WORK

**Temporal Search for Long-Video Understanding.** Traditional video understanding methods rely on static frame sampling, such as uniform sampling or heuristic-based strategies (Li et al., 2024; Chen et al., 2024; Bai et al., 2025), which fail to adapt to varying information density and evolving reasoning contexts. Recent work has explored more sophisticated mechanisms. Similarity-based methods like KeyVideoLLM (Liang et al., 2024) achieve significant compression while maintaining performance , while learning-based approaches such as Frame-Voyager (Yu et al., 2025b) rank frame combinations based on prediction losses, emphasizing task-specific selection. Advanced semantic frameworks have emerged to address temporal dependencies. Logic-in-Frames (Guo et al., 2025) defines logical relations including spatial co-occurrence and temporal proximity to guide dynamic frame sampling. T* (Ye et al., 2025) reframes temporal search as spatial search with adaptive zooming mechanisms. Interactive agents like VideoAgent (Wang et al., 2024) and VideoTree (Wang et al., 2025) enable multi-turn temporal exploration through prompt-driven orchestration. However, none of the aforementioned methods adopt end-to-end optimization, resulting in suboptimal search strategies.

**Reinforcement Learning for Multimodal Reasoning.** Recent advances have explored RL to enhance reasoning capabilities in LLMs. GRPO (DeepSeek-AI, 2025) demonstrates that outcome-based rewards can effectively elicit complex reasoning. Search-R1 (Jin et al., 2025) extends this paradigm to text-based search tasks, showing that RL can facilitate adaptive information retrieval. Approaches like MM-Eureka (Meng et al., 2025) and LMM-R1 (Peng et al., 2025) have successfully applied RL to enhance multimodal reasoning, but focus primarily on static image understanding rather than dynamic video interaction. Video-R1 (Feng et al., 2025) applies GRPO to video reasoning but limits the thinking process to pure text without visual interaction, while DeepEyes (Zheng et al., 2025) uses RL for high-resolution image understanding through adaptive cropping operations but focuses on spatial rather than temporal exploration. Despite these advances, applying RL to interactive long video understanding remains largely unexplored and presents unique challenges.

## 5 CONCLUSION

In this work, we propose TimeSearch-R, a framework that reformulates temporal search as text–video interleaved thinking to learn optimal search strategies directly from data. To enhance temporal search through RL, we propose CSV as a complement to the outcome-only reward of GRPO, addressing the challenges of insufficient temporal exploration and inconsistent logical reasoning. TimeSearch-R achieves strong performance on both temporal search and long-form video understanding tasks, while exhibiting distinct search patterns across different task types. We hope this work contributes meaningful progress toward advancing long video understanding powered by reinforcement learning.

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

# TIMESEARCH-R: ADAPTIVE TEMPORAL SEARCH FOR LONG-FORM VIDEO UNDERSTANDING VIA SELF-VERIFICATION REINFORCEMENT LEARNING

## APPENDIX

This appendix provides more details about our methods, dataset, training, more case studies, broader impacts, as well as the LLM usage, organized as follows:

- Section A: Search Function
- Section B: Dataset Details
- Section C: Prompt Design
- Section D: Evaluation Metrics
- Section E: Efficiency Analysis
- Section F: Search Analysis
- Section G: Training Details
- Section H: More Case Studies
- Section I: Boarder Impacts
- Section J: LLM Usage

## A  SEARCH FUNCTION

### A.1  FRAME SELECTION

The video `search` function selects the most informative frames within predicted temporal clips. Specifically, we leverage determinantal point process (DPP) (Kulesza & Taskar, 2012) as the search optimization for its ability to naturally balance query relevance and diversity that penalizes redundancy, which has been widely applied in information retrieval (Celis et al., 2018; Sun et al., 2025).

Recall the definition of `search` in Sec. 2.1, it aims to select $F$ optimal frames guided by a temporal clip $[t_s, t_e]$ and a query $q$ from the original video $V$. First, the function first subsamples $N$ candidate frames $\mathcal{F}_{[t_s,t_e]} = \{v_i\}_{i=1}^N$ within the temporal clip. Subsequently, we obtain a visual embedding $\mathbf{h}_i \in \mathbb{R}^d$ for each candidate frame in $\mathcal{F}_{[t_s,t_e]}$, and a query embedding $\mathbf{q} \in \mathbb{R}^d$ for $q$. Then we define the pairwise cosine similarity for candidate frames as $S_{ij} = \mathbf{h}_i^\top \mathbf{h}_j$ and compute an unnormalized query relevance score for each frame as $\tilde{r}_i = \mathbf{q}^\top \mathbf{h}_i$, which is rescaled to $[0, 1]$ by min-max normalization $r_i = \frac{\tilde{r}_i - \min \tilde{\mathbf{r}}}{\max \tilde{\mathbf{r}} - \min \tilde{\mathbf{r}} + \epsilon}$, where $\epsilon$ is a small constant to avoid division by zero. The kernel is constructed by diagonal conditioning with these relevance weights:

$$\tilde{\mathbf{L}} = \mathrm{diag}(\mathbf{r})\,\mathbf{S}\,\mathrm{diag}(\mathbf{r}), \tag{6}$$

which is equivalent to $\tilde{L}_{ij} = r_i r_j \mathbf{h}_i^\top \mathbf{h}_j$. The optimal subset $V^* \subset \mathcal{F}_{[t_s,t_e]}$ with $|V^*| = F$ is then obtained through fast greedy MAP inference (Chen et al., 2018):

$$V^* = \arg \max_{S \subseteq \mathcal{F}_{[t_s,t_e]}, |S|=F} \det(\tilde{\mathbf{L}}_S). \tag{7}$$

This formulation ensures that selected frames are both diverse and relevant to the query. When available frames are fewer than $F$, the search function degrades to uniform temporal sampling.

### A.2  FRAME REPRESENTATION

The selected clip frames are sparse and non-uniform. To maintain the temporal pace, we attach an explicit absolute timestamp to each frame by inserting a short text token with the time in seconds (e.g.,

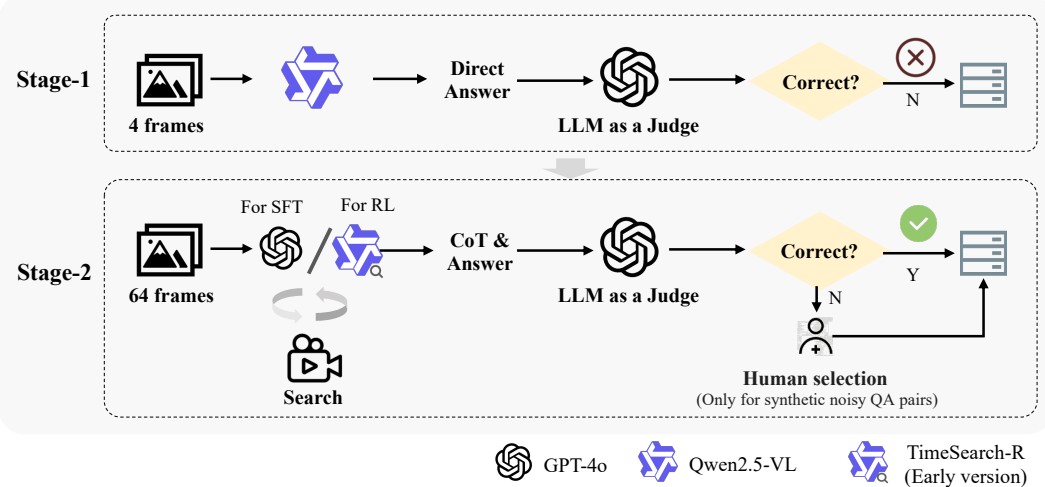

Figure 6: **Illustration of the proposed two-stage data filtering pipeline.**

"12.3s") immediately before the image. This simple interleaving of timestamp text and the corresponding image maintains absolute temporal grounding when inter-frame intervals vary and complements the native temporal ids. Explicit absolute timestamp augmented frame representation has also been observed to improve temporal capability in prior work on long-video temporal grounding (Pan et al., 2025). For uniformly sampled preview frames, we employ the native dynamic-FPS and absolute time encoding following Qwen2.5-VL (Bai et al., 2025), which bind image token sequences to temporal ids aligned with real absolute timestamps.

## B  DATASET DETAILS

### B.1  DATASET CONSTRUCTION

To ensure high-quality training data, we implement a two-stage filtering pipeline as shown in Figure 6.

**Stage 1: Visual Dependency Filtering.** We uniformly sample 4 frames from each video and feed them along with the question to Qwen2.5-VL for inference. Questions that can be correctly answered with this limited visual information are considered to have low visual dependency and are subsequently filtered out. Only questions requiring richer visual context proceed to the next stage.

**Stage 2: Search Usefulness Filtering.** We increase the frame input to up to 64 frames and employ different LVLMs to perform dynamic temporal search for question-relevant video segments. Specifically, we use GPT-4o to generate SFT (Supervised Fine-Tuning) data and an early version of TimeSearch to obtain RL (Reinforcement Learning) training data. Although this stage produces CoT, only the CoT generated by GPT-4o is used for SFT training, while RL training utilizes only the question-answer pairs. To avoid search format errors, we implement format validation for LVLMs' responses, automatically retrying the model until obtaining properly formatted answers.

**Human Selection for VideoMarathon (Panda-70M).** Given that VideoMarathon's training set contains automatically generated question-answer pairs with potential unanswerable questions or incorrect ground-truth answers, we conduct manual annotation to ensure data quality. To minimize annotator bias in model answer evaluation, we establish a structured annotation protocol. First, annotators assess question reasonableness based on video content, filtering out unanswerable or ambiguous questions. Subsequently, annotators provide manual answers and compare them with synthetic ground-truth labels, removing data samples that are inconsistent with human responses.

### B.2  DATASET ANALYSIS

The dataset exhibits a pronounced long-tail distribution in video duration with a mean length of 1,659 seconds. Most videos are shorter than 2,000 seconds, while a nontrivial tail extends beyond one hour,

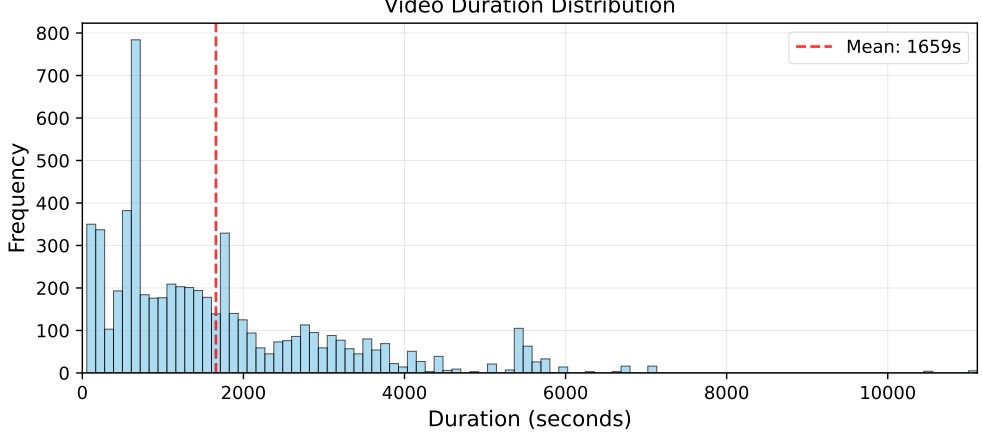

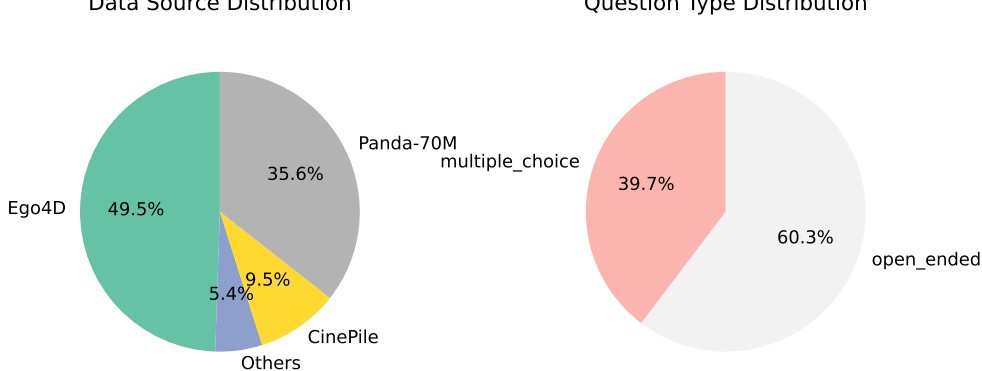

Figure 7: **Dataset analysis.** (1) The training set is mainly composed of long videos. The average length is 1659 seconds, and the maximum length exceeds 10,000 seconds. (2) Egocentric QA pairs come from Haystack-Ego4D, and Exocentric QA data mainly from VideoMarathon and Cinepile, where VideoMarathon employs Panda-70M as the video source. (3) Question types include multiple-choice and open-ended questions. To obtain open-ended QA pairs, we convert some multiple-choice tasks into open-ended questions.

posing significant challenges for static frame sampling. This distribution motivates adaptive temporal search and multi-turn interaction to progressively retrieve evidence under tight keyframe budgets.

We curate data from four major sources to ensure coverage of diverse visual domains and camera styles. As shown in Figure 7, Ego4D from Haystack-Ego4D (Ye et al., 2025) training set contributes 49.5% of samples, providing egocentric daily activities with frequent viewpoint changes. Panda-70M from VideoMarathon (Lin et al., 2025) accounts for 35.6%, expanding the variety of internet videos with heterogeneous motion patterns and scene dynamics. CinePile (Rawal et al., 2024) provides 9.5% of short videos with narrative structure and rapid scene transitions. The remaining 5.4% are from other sources and serve to reduce distributional bias.

Question types are intentionally imbalanced toward open-ended reasoning to better evaluate generative capabilities. Open-ended questions make up 60.3% of the data and emphasize step-by-step analysis, temporal grounding, and explanation quality. Multiple-choice questions comprise 39.7% and offer reliable automatic evaluation signals that complement outcome rewards in RL.

This composition yields wide coverage over motion intensity, scene diversity, and narrative structure while maintaining sufficient automatic evaluability. The mixture of long-tail durations and open-ended questions creates a setting where end-to-end RL and adaptive temporal search offer clear benefits over single-shot heuristics.

## C PROMPT DESIGN

We design prompts to standardize interaction formats, minimize ambiguity, and provide explicit priors for temporal reasoning. Figure 8–11 show the templates used during training and evaluation.

```
                           System Prompt

You are a helpful video assistant.
# Tools
You may call one or more functions to assist with the user query.
You are provided with function signatures within <tools></tools> XML tags:
<tools>
{"type": "function", "function": {"name": "seek_video_frames", "description": "Search and
select video frames according to textual query and temporal window. Time is in seconds.",
"parameters": {"type": "object", "properties": {"query": {"type": "string", "description":
 "The query is used to describe the object, scene, or event of interest in the video
thoroughly and clearly. "}, "start_time": {"type": "number", "description": "Start time of
 the segment of interest. "}, "end_time": {"type": "number", "description": "End time of
the segment of interest. "}, "num_frames": {"type": "integer", "description": "Number of
frames to sample (maximum ${MAX_NUM_FRAMES_PER_TURN}). Default is ${
MAX_NUM_FRAMES_PER_TURN}."}}, "required": ["query"]}}}
</tools>
For each function call, return a json object with function name and arguments within <
tool_call></tool_call> XML tags:
<tool_call>
{"name": <function-name>, "arguments": <args-json-object>}
</tool_call>
```

Figure 8: **The system prompt with temporal search tools.**

**System Prompt.** We follow the tool-use specification of the base Qwen2.5-VL family (Bai et al., 2025) and adopt its `tool_call` schema for invoking temporal search. The hyperparameter `MAX_NUM_FRAMES_PER_TURN` in Figure 8 controls the maximum number of frames per search turn, while `NUM_FRAMES_PREVIEW` specifies the number of preview frames provided initially. During training, `MAX_NUM_FRAMES_PER_TURN` is set to 8 by default. At test time, we adjust both parameters to control the target average frame count. Specifically, since the model typically performs ∼1.2 search turns beyond the initial preview, setting `NUM_FRAMES_PREVIEW=4` and `MAX_NUM_FRAMES_PER_TURN=3` yields an average of ∼7.8 frames. Similarly, we set `NUM_FRAMES_PREVIEW=758` and `MAX_NUM_FRAMES_PER_TURN=8` for the 768-frame setting. The frame numbers in the tables of main text represent actual statistics averages across all test samples after the inference is completed.

**Question Answering Prompt.** The QA template enforces thorough reasoning inside `<think>` before any tool call or final answer. It restricts the output to exactly one of two formats and allows at most eight rounds of `<tool_call>`. It explicitly provides the line *"The video duration: {duration} seconds."* to help the model produce absolute timestamps better. See Figure 9.

```
                         Question Answering

You must ALWAYS conduct thorough reasoning inside <think> and </think> tags BEFORE calling
 any tool or answering the question.
You must invoke tools to explore any video content you are interested in within <tool_call
> </tool_call> tags.
You are allowed to use <tool_call></tool_call> tags for a maximum of 8 rounds.
When you have enough information to answer the question, provide your answer within <
answer> </answer> tags. Your answer should be supported by evidence from the video.
Your output must follow the format: <think>Your reasoning process</think><tool_call>
Parameters</tool_call> or <think>Your reasoning process</think><answer>Your answer</answer
>Question: {question}
The video duration: {duration} seconds.
```

Figure 9: **The template for question answering.**

**Clip Frame Sampling and Search Response.**   After a search, the template returns the selected frames and their corresponding timestamps. If the frames are sufficient, the model must place the final answer in `<answer>`. Otherwise, the template asks the model to call the tool again with different parameters in JSON, thereby encouraging reflection and re-query. See Figure 10.

---
**Temporal Search Response**

```
Here are selected frames. They are located at {timestamps}.
If the frames provided above are sufficient to answer the user's question, please put your
 final answer within <answer></answer>.
Otherwise invoke the tool again with different parameters in JSON format.
```
---

Figure 10: **The response template of the temporal search.**

**Completeness Self-Verification Prompt.**   The CSV template asks the model to answer as briefly as possible and to say *"I don't know"* when the visual evidence is insufficient. No tools are available in this stage, which prevents new searches and ensures the answer is grounded only on the dynamic frame set gathered earlier. See Figure 11.

---
**Completeness Self-Verification**

```
You are a helpful assistant. Please answer visual questions as briefly as possible. When
you don't have enough visual information, please say 'I don't know'.
```
---

Figure 11: **The template for CSV reasoning.**

# D    EVALUATION METRICS

**Completeness Rate.**   We measure the proportion of cases where the dynamic visual context alone suffices to produce the correct answer. Concretely, after the multi-turn search, we re-answer the question using only the gathered dynamic frame set and disallow further search, following the CSV procedure in Sec. 2.2 and the prompt illustrated in Figure 11. The resulting correctness is computed with the same task-specific accuracy used elsewhere, averaged over the whole dataset.

**Consistency Rate.**   Consistency evaluates whether the intermediate reasoning coherently supports the final answer under the given question. We prompt a LLM model (GPT-4o) with the question, the reasoning text extracted from `<think>...</think>`, and the final answer from `<answer>...</answer>`, using the format in Figure 12 that requires a structured output: a short analysis in `<think>` followed by `<answer>` equal to "Yes" or "No". In implementation, we parse the LLM's output to obtain the binary decision; "Yes" is counted as 1 and "No" as 0, and any parsing failure is treated as 0. The Consistency Rate is the dataset average of these binary outcomes.

---
**Consistency Score Evaluation**

```
<system prompt>
You are a careful and logical reviewer. Your task is to verify whether the given reasoning
 process and the final answer are consistent in addressing the given question.

Please carefully read the following information:

Question: <Question>
Reasoning Process: <Reasoning>
Final Answer: <Answer>

Please follow this format strictly:
<think> Your analysis here </think> <answer> Yes/No </answer>
```
---

Figure 12: **The template for calculating consistency.**

# E  EFFICIENCY ANALYSIS

## E.1  TRAINING EFFICIENCY

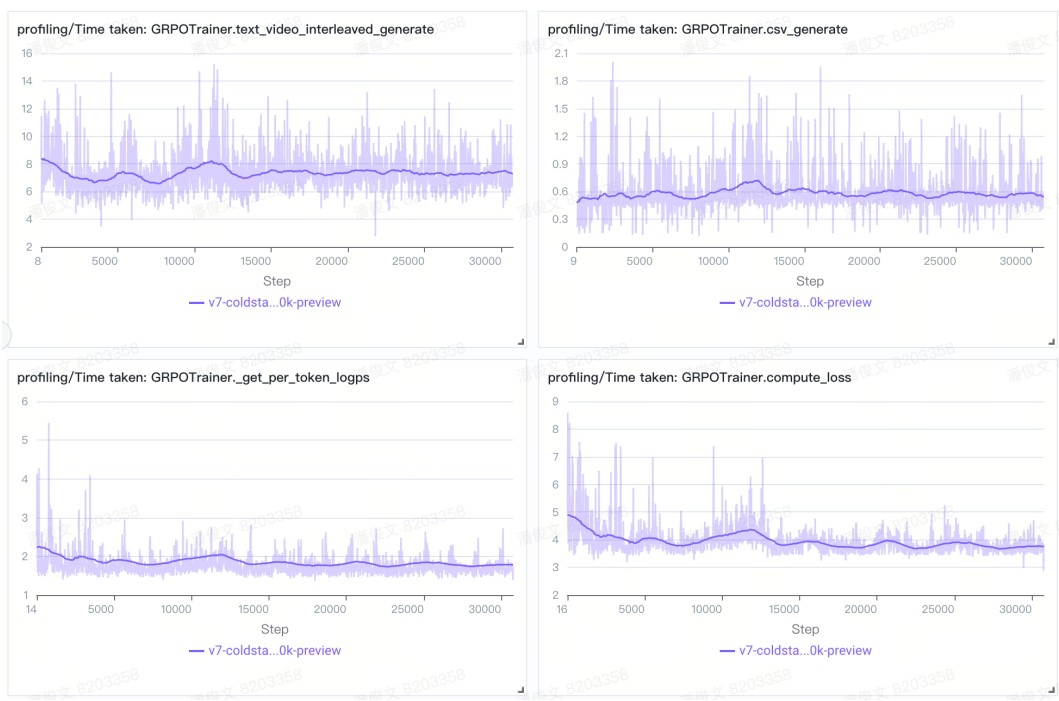

Figure 13: **Training efficiency of CSV.** CSV phase requires 0.6 seconds on average, which is substantially lower than the complete rollout (7.5 seconds).

We conduct a comprehensive profiling analysis to evaluate the training efficiency, specifically examining the overhead introduced by the CSV phase. As illustrated in Figure 13, the CSV phase requires 0.6 seconds on average, which is substantially lower than the time required for a complete text-video interleaved rollout (7.5 seconds). Our profiling also covers other components, including per-token log probability calculation and loss computation. Overall, the average training time is 56.09 seconds per iteration. The additional overhead from the CSV phase is minimal, peaking at 2.1 seconds with an average of 0.6 seconds. This accounts for approximately 1.1% of the total training duration, demonstrating that the impact on overall training efficiency is negligible.

## E.2  INFERENCE LATENCY

Table 5: **End-to-end inference efficiency.** We report the search model, number of frames, search/overall latency, number of LLM turns, VideoMME performance, and temporal/visual F1 scores.

| Method | Search Model | # Frame | Search (sec) | Overall (sec) | # LLM Turns | VideoMME ↑ | Temporal $F_1$ ↑ | Visual $F_1$ ↑ |
|---|---|---|---|---|---|---|---|---|
| Qwen2.5-VL-7B | N/A | 8 | N/A | 4.02 | 1 | 54.7 | - | - |
| T*+GPT-4o | YOLO-world-110M×49 | 8 | 6.4 | 26.00 | 2 | 56.5 | 2.5 | 64.3 |
| TimeSearch-R | SigLIP-400M×2.3 | 7.8 | 0.138 | 4.65 | 2.3 | 59.1 | **8.1** | **69.2** |
| TimeSearch-R + GPT-4o | N/A | 7.8 | N/A | 12.55 | 3.3 | **68.5** | - | - |

As shown in Table 5, we evaluate the inference efficiency of different methods on the VideoMME benchmark using NVIDIA A100-80GB GPUs. T*+GPT-4o requires 26.00 seconds per sample, with the question grounding, iterative search, and question answering stages taking 11.7 s, 6.4 s, and 7.9 s, respectively. In contrast, TimeSearch-R achieves a significantly faster end-to-end latency of only 4.65 seconds per sample, using an average of 7.8 frames and 2.3 LLM turns. Benefiting from the compact 7B backbone, TimeSearch-R ensures superior inference efficiency while achieving better performance (59.1 vs. 56.5 on VideoMME). The efficiency advantage of TimeSearch-R stems from three key

factors. First, thanks to modern LLM infrastructure with KV-cache optimization, TimeSearch-R's multi-turn dialogue (2.3 turns) incurs only minimal additional overhead compared to single-turn inference of Qwen2.5-VL. Second, video frame features only need to be computed once and can be cached across multiple dialogue turns, with this computation parallelizable with the LLM's prefilling phase. We utilize an industrial-level deployment mechanism (clip-as-service) to further accelerate SigLIP computation. Third, the temporal search operation itself is extremely lightweight, taking only 0.138 seconds per sample (less than 3% of the total latency), as it comprises only text embedding of queries and feature-based retrieval operations. Meanwhile, TimeSearch-R achieves substantially better temporal localization accuracy, with a Temporal $F_1$ of 8.1 compared to 2.5 for T*+GPT-4o, underscoring the effectiveness of our reinforcement-driven search policies.

# F  SEARCH ANALYSIS

## F.1  DISTRIBUTION OF FRAME NUMBERS AND SEARCH TURNS

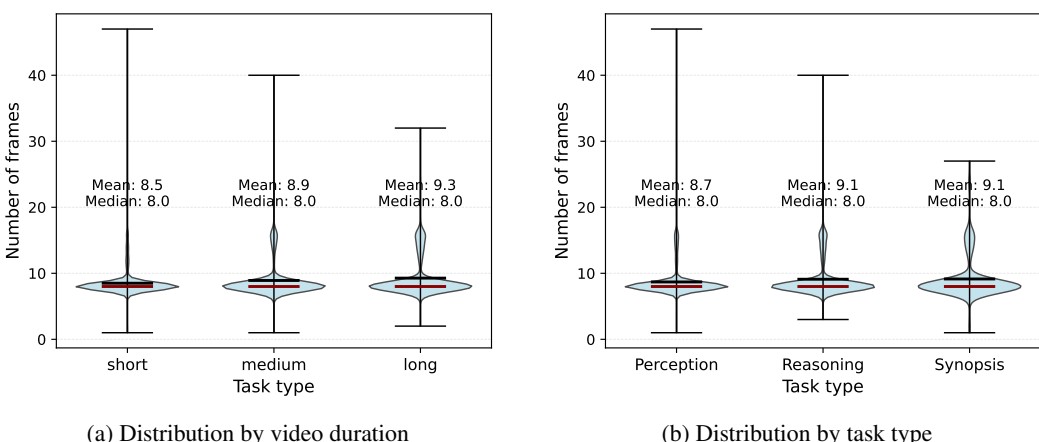

(a) Distribution by video duration      (b) Distribution by task type

Figure 14: **Number of selected frames distribution.** We visualize the distribution of sampled frames across different video durations (a) and task types (b) on VideoMME.

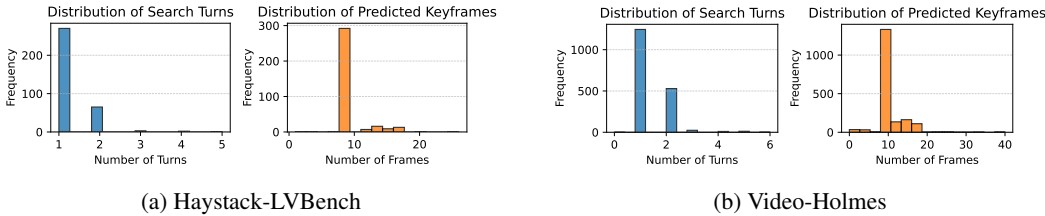

(a) Haystack-LVBench      (b) Video-Holmes

Figure 15: **Number of search turns and frames distribution.** We visualize the distribution for video understanding task Haystack-LVBench (a) and video reasoning task Video-Holmes (b).

We visualize the distribution of selected frames across different video durations and task types on VideoMME in Figure 14. We observe that (1) longer videos have more average frames, and (2) more complex reasoning tasks and comprehensive tasks obtain more frames. We further visualize the detailed distribution of search turns and frames on Haystack-LVBench for long-form video tasks and Video-Holmes for complex video reasoning tasks in Figure 15. Most questions in Haystack-LVBench and Video-Holmes require only one or two search turns to obtain the correct answer.

## F.2  DIAGNOSTIC BENCHMARK ANALYSIS.

To better understand the specific capabilities improved by TimeSearch-R, we conduct diagnostic analysis using VideoEvalPro (Ma et al., 2025) and VideoMME (Ye et al., 2025), which categorize video understanding tasks into different dimensions. VideoEvalPro distinguishes between local and holistic

Table 6: **Diagnostic performance on VideoEvalPro**. TimeSearch-R significantly improves the performance on local understanding tasks.

| Models | #Frames | Local | Holistic | Overall |
|---|---|---|---|---|
| Qwen2.5-VL | 512 | 51.1 | 37.2 | 46.9 |
| TimeSearch-R | 512 | $55.8_{+4.7}$ | $35.8_{-1.4}$ | $49.9_{+3.0}$ |

Table 7: **Diagnostic performance on VideoMME.** We report the detailed perception and reasoning performance on VideoMME. TimeSearch-R significantly improves reasoning performance.

| Model | Perception Details | | | | | | | Reasoning Details | | | | Synopsis | Perception | Reasoning |
|---|---|---|---|---|---|---|---|---|---|---|---|---|---|---|
| | Temporal | Spatial | Attribute | Action | Object | OCR | Counting | Temporal | Spatial | Action | Object | | | |
| Qwen2.5-VL-7B | 80.0 | 61.1 | 77.5 | 66.1 | 67.6 | 71.9 | 44.8 | 51.4 | 76.8 | 56.8 | 59.5 | 80.9 | 65.1 | 58.2 |
| TimeSearch-R-7B | 72.7 | 66.7 | 77.9 | 66.8 | 73.2 | 74.1 | 43.7 | 58.8 | 75.0 | 62.5 | 61.9 | $79.6_{-1.3}$ | $66.7_{+1.6}$ | $62.2_{+4.0}$ |

understanding, while VideoMME separates tasks into perception, reasoning, and synopsis categories. Table 6 shows that TimeSearch-R achieves substantial improvements on local understanding tasks (+4.7), leading to an overall gain of +3.0 points. Table 7 reveals consistent enhancements in reasoning capabilities (+4.0), with the most significant improvement observed in temporal reasoning (+7.4). These results indicate that TimeSearch-R particularly excels at tasks requiring fine-grained temporal analysis and complex reasoning over video content. The improvements are attributed to the learned search strategies, which adaptively selects task-relevant frames through multiple search turns. This dynamic frame retrieval enables the model to capture precise spatiotemporal details critical for local understanding and temporal reasoning, rather than relying on uniform sampling of the entire video.

# G  TRAINING DETAILS

Table 8: **Training hyperparameters of TimeSearch-R.**

| Category | Parameter | Value |
|---|---|---|
| **Video Processing** | Max FPS | 2 |
| | Max Frames per Video | 768 |
| | Total Video Tokens | 10,240 |
| | Min Tokens per Frame | 12 |
| | Max Tokens per Frame | 256 |
| **Interaction Settings** | Max Search Turns | 8 |
| | Max Completion Length per Turn | 256 |
| **GRPO Training** | Number of Generations | 8 |
| | KL Penalty Coefficient ($\beta$) | 0.005 |
| | Scale Rewards | false |
| | Batch Size per GPU | 1 |
| | Gradient Accumulation Steps | 2 |
| **Infrastructure** | DeepSpeed Configuration | ZeRO-3 Offload |
| | VLLM Mode | colocate |
| | Replay Buffer | true |

We summarize the key hyperparameters in Table 8 for reproducibility.

**Training Configuration.** TimeSearch-R employs a distributed training setup using PyTorch's native distributed data parallel framework with ZeRO-3 memory optimization through DeepSpeed. The training process leverages gradient accumulation to simulate larger batch sizes while maintaining memory efficiency on GPU clusters. We utilize mixed precision training with bfloat16 to accelerate computation while preserving numerical stability, coupled with Flash Attention 2.0 for efficient attention computation.

**GRPO Training Setup.** The reinforcement learning phase uses Group Relative Policy Optimization with 8 generations per prompt to provide sufficient policy gradient estimates. The KL divergence penalty coefficient $\beta$ is set to 0.005 to balance between reward optimization and policy regularization. We employ VLLM in colocate mode for efficient inference during rollout generation, enabling faster policy updates. This RL training stage is implemented on top of the TRL library (von Werra et al., 2020), following standard practice for outcome-driven policy optimization in large language models.

**Video Processing Configuration.** The model processes videos with a maximum of 768 frames and allocates up to 10,240 tokens for video content representation. Each interaction turn is limited to 8 search operations, with a maximum of 8 interaction turns per question to ensure comprehensive temporal exploration while maintaining computational efficiency. Frame tokens are dynamically allocated between 12 and 256 tokens per frame based on content complexity and relevance.

## H   MORE CASE STUDIES

This section provides more case studies of TimeSearch-R, including successful cases and failed cases.

**Successful Cases.** These representative success cases illustrate how TimeSearch-R conducts multi-turn exploration to accumulate decisive visual evidence while maintaining alignment between the reasoning trace and the final answer. They encompass confirmation (Figure 16), elimination (Figure 17), and sequential exploration patterns (Figure 18), collectively demonstrating that the policy preserves the high completeness and consistency reported in Sec. D.

**Failed Cases.** Figure 19 illustrates a failure where the model generates repetitive reasoning that exhausts the search budget. Figure 20 illustrates a failure where the model makes temporal prediction errors, such as outputting timestamps beyond the video duration. Figure 21, 22, and 23 illustrate failures involving incomplete, inconsistent, or hallucinated reasoning. Figure 24 illustrates a failure where an excessive frame budget leads the model to retrieve irrelevant frames.

## I   BROADER IMPACTS

TimeSearch-R contributes to several important areas beyond the immediate technical contributions:

**Advancing Video Interpretability and Explainability.** TimeSearch-R introduces interleaved text–video reasoning traces that provide transparent insights into the model's decision-making process. The completeness and consistency criteria we propose enable quantitative assessment of long-form video explanations, making temporal search decisions auditable and interpretable. This advancement represents a significant step toward more explainable AI systems in the video domain, where understanding the reasoning process is crucial for building trust and ensuring reliability.

**Transforming Video Reasoning from Static to Dynamic Paradigms.** Our approach fundamentally shifts the paradigm from static frame sampling to dynamic, interactive reasoning in video understanding. By operationalizing hypothesis-driven exploration through iterative temporal search, we promote a new methodology that emphasizes transparent, stepwise evidence gathering. This contrasts sharply with traditional one-shot inference over fixed visual contexts, encouraging researchers to develop more adaptive and interactive AI systems. The demonstrated effectiveness of our approach may inspire broader adoption of similar interactive paradigms across various multimodal tasks.

**Exploring Scalable Weakly-Supervised Process Rewards.** We introduce outcome-based process supervision that eliminates the need for costly process annotations. Through the integration of weak supervision and reinforcement learning via completeness self-verification, our method successfully aligns intermediate search decisions with correct outcomes. This approach offers a scalable solution for training complex interactive systems without requiring fine-grained procedural labels, potentially reducing annotation costs and enabling broader application across diverse domains.

## J  THE USE OF LARGE LANGUAGE MODELS

The authors declare that Large Language Models were used in this paper for polishing the writing. Specifically, the LLM assisted with tasks such as grammar checking, sentence simplification, and improving the overall fluency of the text. It is important to note that the LLM was not used for any literature review or research ideation. All research ideas and experimental analyses presented in the paper were solely conducted by the authors.

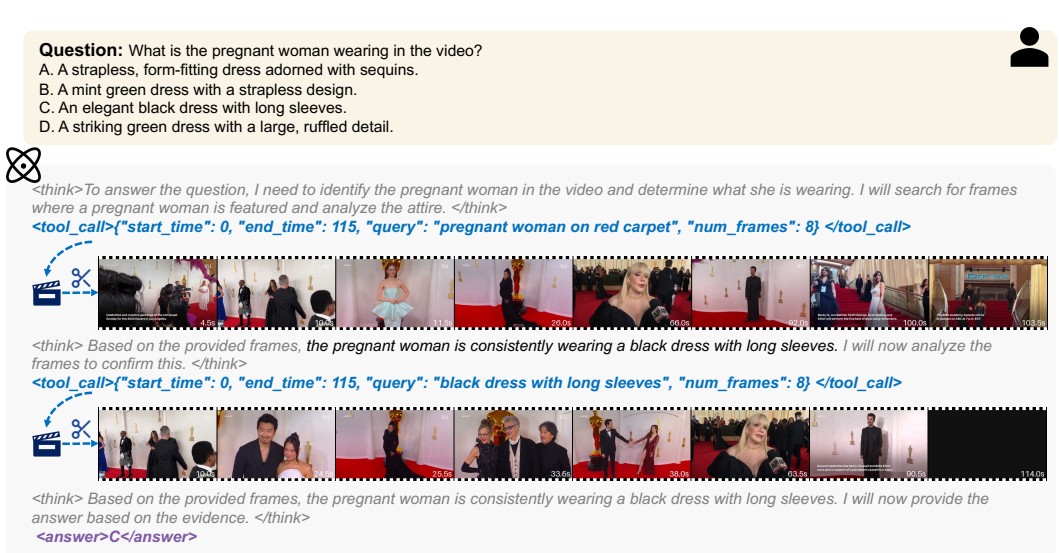

Figure 16: **Search pattern: search confirmation.**

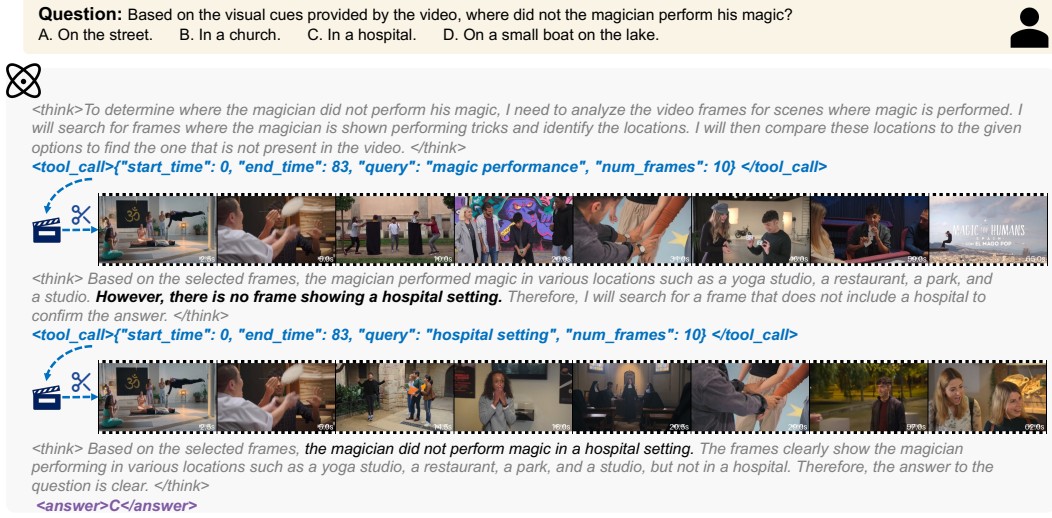

Figure 17: **Search pattern: elimination method.**

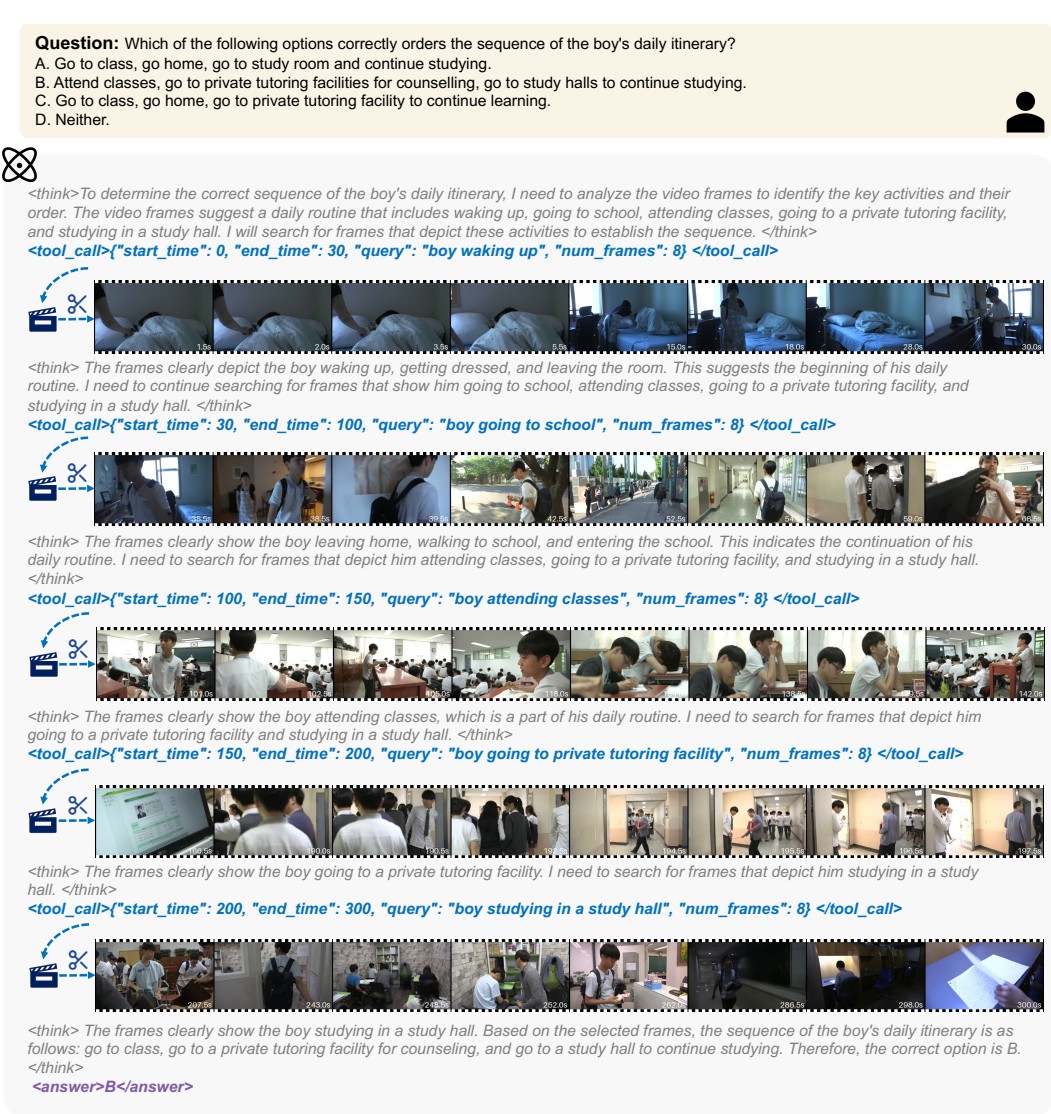

Figure 18: **Search pattern: sequential search.**

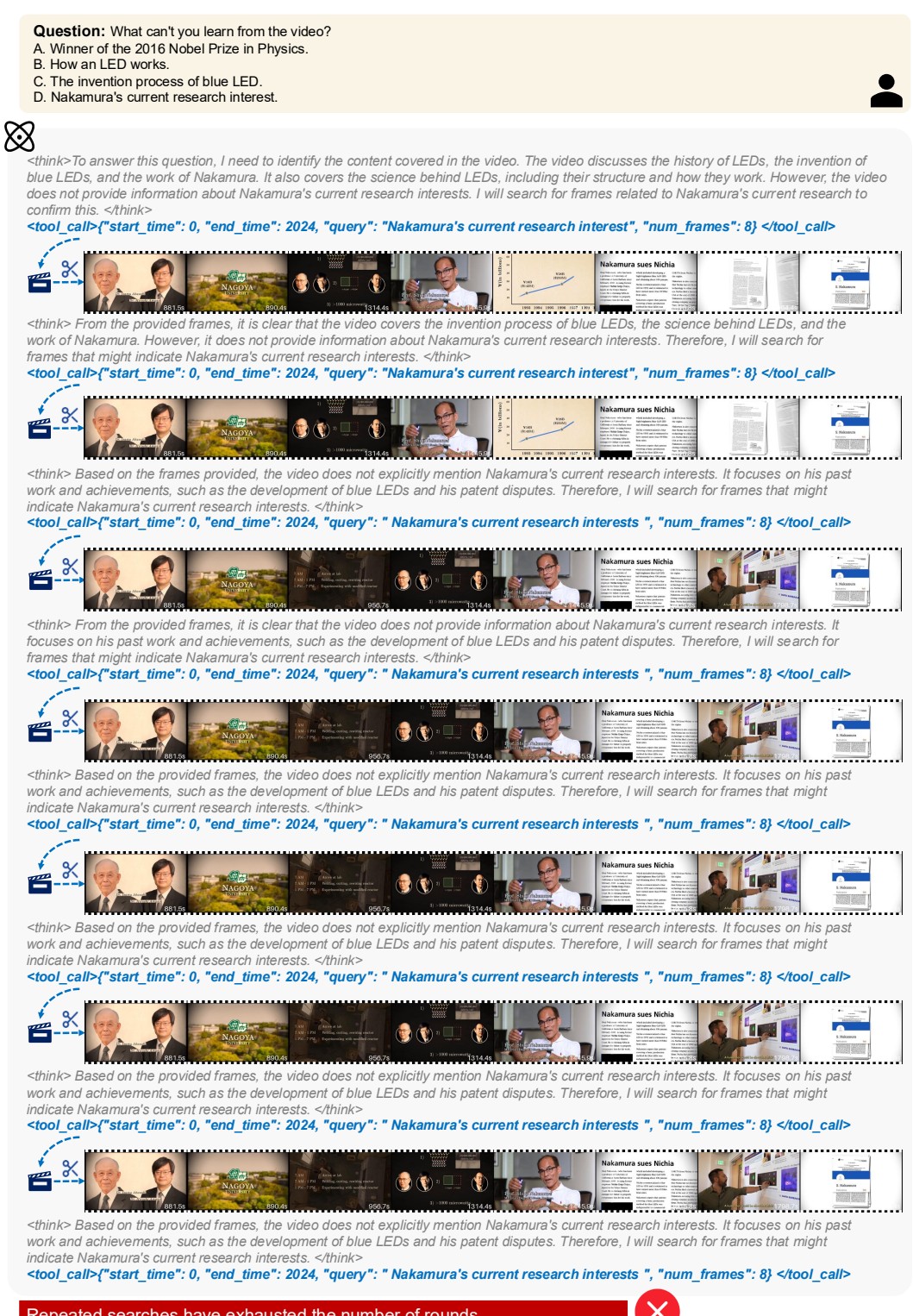

Figure 19: **Failure case: repeated and exhausting search.** The model repeatedly searches for the same information until the maximum number of search turns is reached. This case is rare in our experiments (1% of VideoMME, 0.4% of LongVideoBench, and 0.1% of MLVU).

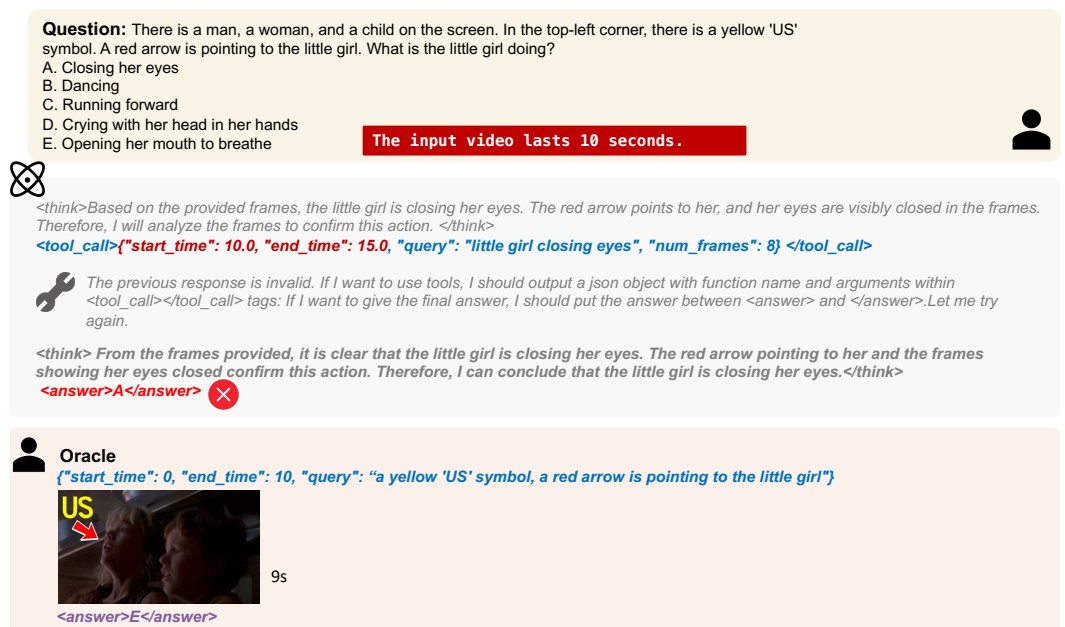

Figure 20: **Failure case: error time parameters.** The predicted start time exceeds the video duration, leading to unexpected search results and an incorrect answer.

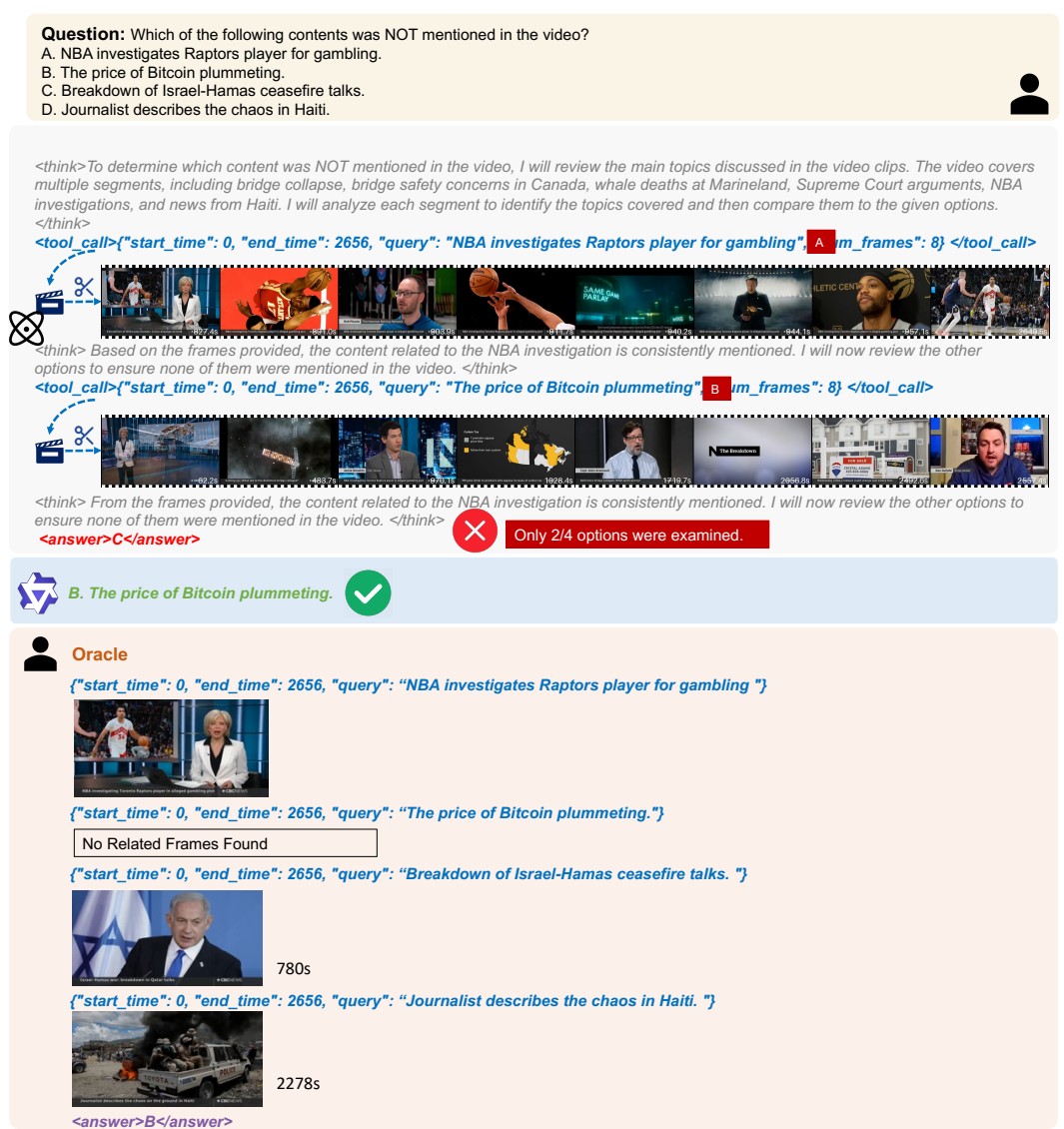

Figure 21: **Failure case: insufficient search.** There were 4 options in total, but only 2 were reviewed before the search was terminated.

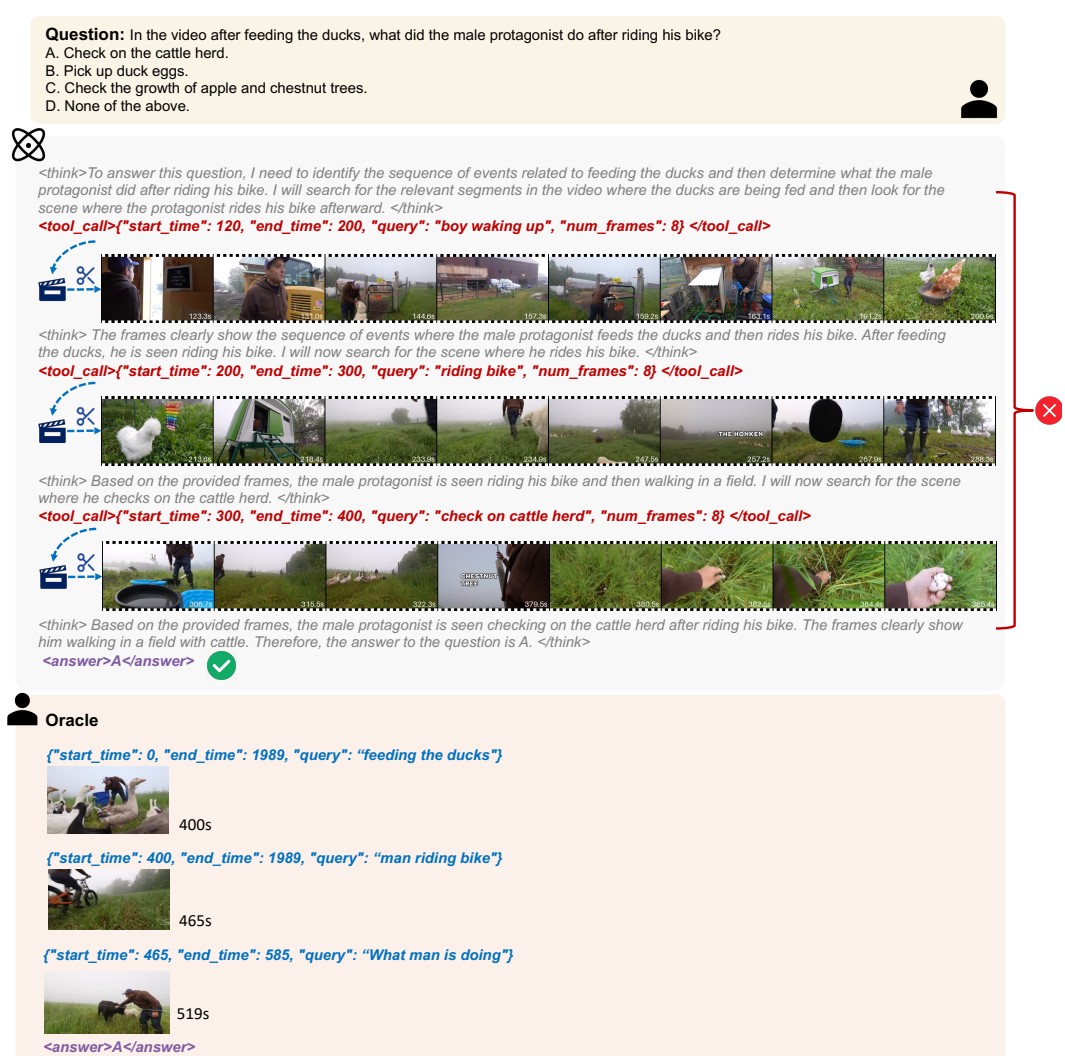

Figure 22: **Failure case: visual hallucination.** No information related to riding was found in the search results.

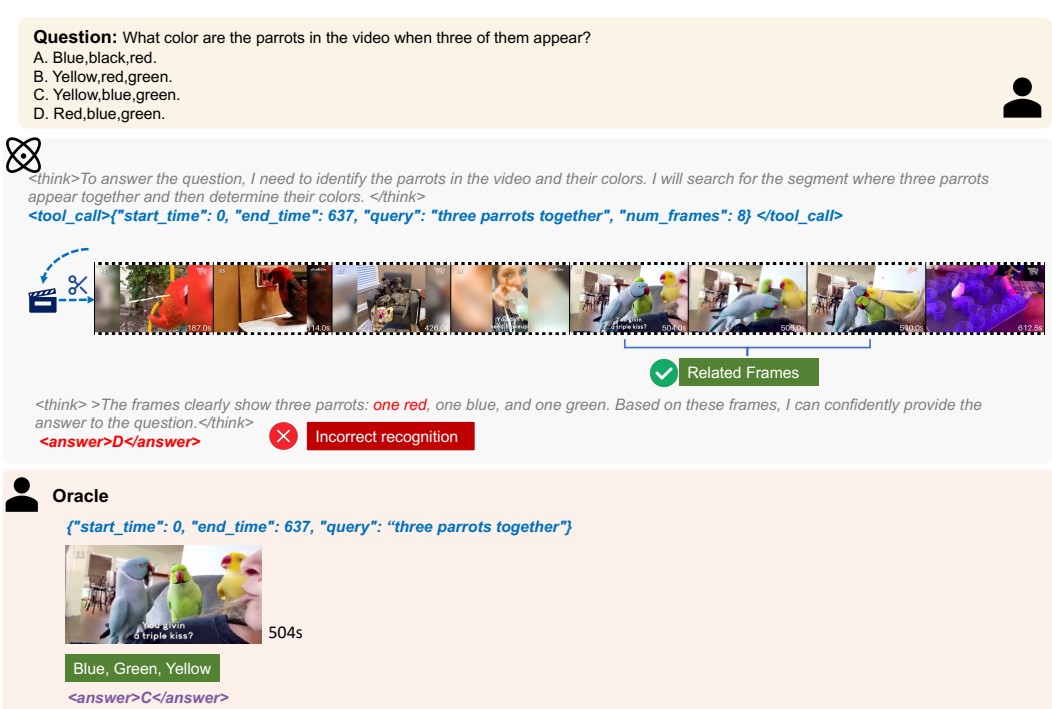

Figure 23: **Failure case: correct frame set but incorrect answer.** The model correctly sampled the frame set but perception or reasoning capabilities are insufficient, leading to an incorrect answer.

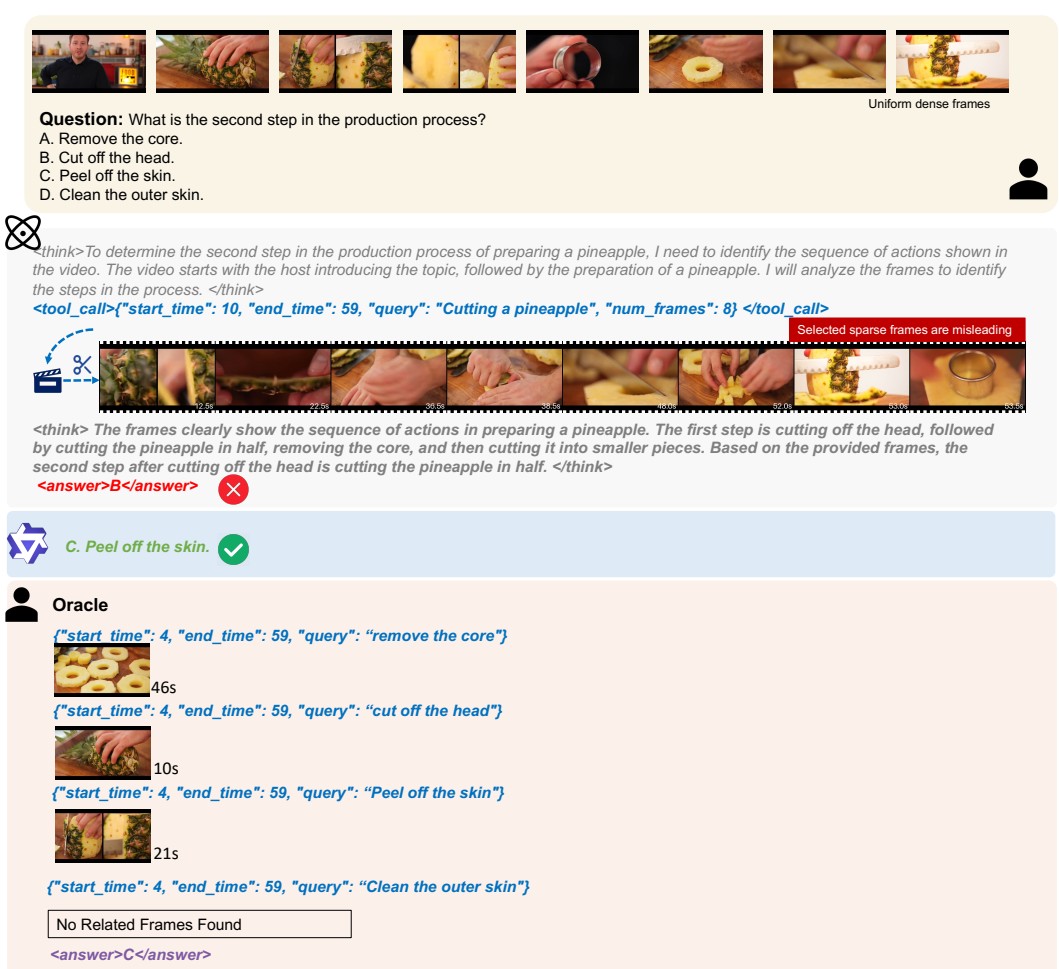

Figure 24: **Failure case: misleading search result.** Due to the limited number of frames in a single search, the sparse frames obtained by the model may contain misleading information, leading to an incorrect answer.

