# OpenReview forum: "TimeSearch-R: Adaptive Temporal Search for Long-Form Video Understanding via Self-Verification Reinforcement Learning"
_ICLR.cc/2026/Conference — ICLR 2026 Poster_

### Official Review · Reviewer_VVki · 2025-10-16

**Soundness:** 3
**Presentation:** 3
**Contribution:** 2
**Rating:** 4
**Confidence:** 4

**Summary:**

This paper tackles the challenge of temporal search in long-form video understanding—how to efficiently identify a small subset of relevant frames among tens of thousands to answer video-based questions. Existing works depend on hand-crafted multi-step pipelines, whereas **TimeSearch-R** proposes an end-to-end reinforcement learning (RL) formulation that interleaves text-video reasoning and frame retrieval. The key technical contribution is GRPO-CSV (Group-Relative Policy Optimization with Completeness Self-Verification), which supplements the standard RL outcome reward with an additional self-verification phase: the model must re-answer the question using only the frames it has searched, thereby improving temporal completeness and reasoning consistency. The paper further introduces a two-stage data filtering pipeline to remove linguistically solvable or unsolvable samples. Empirically, TimeSearch-R achieves new state-of-the-art (SOTA) results on Haystack-LVBench, Ego4D, and long-video reasoning benchmarks, including VideoMME, MLVU, and LongVideoBench, outperforming both open-source and closed-source models. Ablations show CSV prevents RL collapse and improves search completeness.

**Strengths:**

- **Clear Motivation.** Addresses the key limitation of static frame sampling with a well-motivated reformulation of temporal search as text-video interleaved reasoning. Well-written and logically structured, making ideas easy to follow.
- **Algorithmic Novelty.** Proposes GRPO-CSV, a simple yet effective reinforcement-learning mechanism that improves search completeness and reasoning consistency.
- **Remarkable Empirical Results.** Achieves new SOTA on multiple long-video benchmarks with thorough ablations and comparisons.
- **Insightful Analysis.** Provides clear case studies and ablations demonstrating stability gains and human-like search behavior.

**Weaknesses:**

- **Incremental novelty.** While the integration of self-verification into GRPO is interesting, the idea remains an incremental extension of existing RL-based reasoning frameworks, rather than introducing a fundamentally new optimization principle.
- **Limited theoretical justification.** The completeness reward (Eq. 4) is heuristic and may become unstable if the base answer is wrong; no theoretical analysis of convergence or variance reduction is provided.
- **Heuristic parameterization.** Several hyperparameters (e.g., search budget, reward weights, KL penalty) are empirically tuned without sensitivity analysis, leaving uncertainty about robustness across tasks and video lengths.
- **Marginal quantitative gains.** Although the model achieves clear improvements, the gains are moderate on some benchmarks and may not fully justify the added complexity.
- **Scalability and Cost Concerns** Training requires 32 × A100 GPUs with multiple rollouts per prompt. The scalability to larger backbones and the claimed efficiency remain unverified due to missing runtime or complexity analysis.
- **Missing qualitative evaluation.** While one qualitative case is shown, the paper lacks broader visualizations of temporal search evolution or failure cases to illustrate where TimeSearch-R surpasses prior methods.

**Questions:**

- Could the authors clarify how Completeness Self-Verification (CSV) is triggered during training — is it applied to every rollout or only when the original answer passes a correctness threshold?
- How are the reward weights (completeness / format / accuracy) combined or tuned? Would alternative weighting (e.g., annealing completeness) affect training stability?
- Could the authors quantify the number of samples filtered at each stage and analyze its impact on performance and generalization?
- Could the authors report variance or confidence intervals over multiple runs to demonstrate result robustness?
- Could the authors provide statistics on average search turns and retrieved frames compared to baseline methods?
- How does the inference time (or throughput in videos/sec) compare to baselines?
- What are the training GPU hours or per-iteration cost introduced by the CSV phase?
- Could the authors provide examples or analysis of typical CSV failure cases (e.g., incomplete retrieval or reasoning drift)?

---

> ### Author Response · Authors · 2025-11-27
> **Response to Reviewer VVki**
>
> The authors thank the reviewer for the detailed evaluation and insightful questions, and provide point-by-point responses as follows:
> ### **1. Method Novelty and Theoretical Analysis**
> We respectfully clarify that our proposed GRPO-CSV is not an incremental extension of existing RL-based reasoning frameworks. Instead, it introduces a novel design that directly addresses a fundamental challenge in long-video understanding and RL reasoning—insufficient supervision of intermediate reasoning steps. First, the completeness reward is not heuristic. It explicitly models the objective of maximizing the mutual information between the video input and final answer. As stated in **L213–215** of the paper, the reward is activated *only when the original answer is correct*, ensuring that it does not introduce instability when the base answer is wrong. We have added additional theoretical analysis in **Section 2.2**:
> * From an information-theoretic perspective, CSV increases the mutual information between video inputs and answers, encouraging deeper utilization of the visual evidence.
> * From an optimization perspective, CSV transforms sparse accuracy rewards into denser signals, enabling better credit assignment and more effective policy optimization.
> ### **2. Hyperparameter Sensitivity Analysis**
> **Reward Weights**
>
> In the main experiments, we set the same weights for completeness, format, and accuracy rewards by default. **Section 3.3** provides a binary ablation of completeness reward. We additionally perform finer-grained ablations over the relative weighting between completeness and accuracy rewards. As shown in the table below, the model is not sensitive to reward weights. Removing the completeness reward reduces search completeness, while including it stabilizes training. The two rewards complement each other, with the optimal ratio lying within [0.25, 0.5].
>
> |$\lambda_{comp/acc}$|Completeness|Accuracy|
> |-|:-:|:-:|
> |SFT|60.5|59.2|
> |0.00|57.2|65.1|
> |0.25|61.5|65.4|
> |0.50|60.2|66.6|
> |0.75|61.5|64.5|
> |1.00|61.2|64.8|
>
> **KL Penalty**
>
> We also ablate the KL-penalty coefficient with results shown below. While KL regularization helps maintain training stability, the model is not sensitive to its precise value. We use 0.005 for all main experiments.
>
> |$\lambda_{KL}$|Completeness|Accuracy|
> |-|:-:|:-:|
> |0|-|collapse|
> |0.001|60.3|65.4|
> |0.005|60.2|66.6|
> |0.01|60.9|66.5|
> |0.05|61.9|64.9|
> ### **3. Data Filtering Pipeline**
> We employ a two-stage filtering pipeline to construct a high-quality training dataset. The number of samples filtered at each stage is provided in the table below. As described in **Section 2.3**, Stage 1 removes overly simple samples to reduce reliance on linguistic shortcuts. Stage 2 removes overly difficult samples to ensure that the model receives meaningful rewards when actively performing temporal exploration.
>
> |Source|Question type|Original|Stage 1|Stage 2|
> |-|:-:|:-:|:-:|:-:|
> |Haystack-Ego4D-Train|MC&OE|25.8K|25.8K|3K|
> |VideoMarathon (Panda-70M)|OE|1.73M|100K|2K|
> |CinePile|MC|303K|1K|0.5K|
> ### **4. Statistical Robustness**
> Following prior work, inference is performed with temperature = 0 and greedy search sampling, and the batch size is fixed to 1. Therefore, the reported results contain no randomness.
> ### **5. Distribution of Search Turns and Frames**
> We provide the distributions of search turns and retrieved frames in **Appendix Figures 14 and 15**. Most questions require only one or two search turns. Longer videos and more complex reasoning tasks require more frames, indicating that the model has learned adaptive search behavior conditioned on video length and task complexity.
> ### **6. Inference Efficiency Analysis**
> **Appendix Table 5** presents detailed inference efficiency analysis. Compared with the base model, TimeSearch-R adds only 16% inference latency (4.65 vs. 4.02 sec/video) while achieving 4.4% improvement on VideoMME. Compared with T*, TimeSearch-R is dramatically more efficient (4.65 vs. 26.00 sec/video) while achieving stronger performance. When transferring TimeSearch-R’s policy to GPT-4o, performance improves by 9.4%, while total latency remains less than half of T*, demonstrating the efficiency of our search strategy.
> ### **7. Training Efficiency of CSV**
> As shown in **Appendix Figure 13**, each CSV iteration takes only 0.6s, far lower than a full text–video interleaved rollout (7.5s). The CSV stage accounts for only 1.1% of total training time, confirming its training efficiency.
> ### **8. Failure Case Studies**
> The original **Appendix H** provides analyses of emergent search patterns and failure cases. We have now added more failure cases in **Appendix Figures 19–24** and conducted deeper analyses **in Section 3.4** to better illustrate current limitations and highlight directions for future research.
>
> We hope the above responses adequately address the reviewer’s concerns. If the reviewers have any other questions, we would be very glad to have further discussions.

---

### Official Review · Reviewer_mLMd · 2025-10-30

**Soundness:** 3
**Presentation:** 3
**Contribution:** 3
**Rating:** 8
**Confidence:** 4

**Summary:**

This paper presents TimeSearch-R, a novel framework that reformulates temporal search in long-form video understanding as an interleaved text-video thinking process, optimized end-to-end with reinforcement learning. The key innovation is the Completeness Self-Verification (CSV) mechanism, which addresses critical failure modes (insufficient exploration, inconsistent reasoning) in standard outcome-based RL by forcing the model to re-answer questions using only its dynamically retrieved frames. The authors also construct a high-quality dataset via a two-stage filtering pipeline to eliminate samples solvable by linguistic bias or unsolvable even with ideal search. Extensive experiments show that TimeSearch-R achieves new state-of-the-art results on temporal search benchmarks (e.g., Haystack-LVBench) and long-video QA benchmarks (e.g., LongVideoBench), while also exhibiting emergent, human-like search strategies.

**Strengths:**

1. Clear motivation and problem formulation. The paper correctly identifies the bottleneck of current Video-LLMs and reformulates temporal search as an interactive reasoning process.

2. The GRPO-CSV algorithm introduces self-verification to provide intermediate supervision within reinforcement learning.

3. The proposed model shows consistent gains across several long-video understanding benchmarks and produces explicit reasoning–search traces, enhancing interpretability over prior approaches.

**Weaknesses:**

The main concern lies in model coupling and generalization.
It remains unclear whether the learned TimeSearch-R policy is specific to the base VLM (Qwen2.5-VL-7B) or transferable to other model sizes and architectures.
Since the reinforcement training is done with a fixed backbone, it is possible that the learned policy overfits to the internal embedding or feature distribution of that model.
This would limit the practical usefulness of TimeSearch-R as a general temporal search module for other VLMs (e.g., 3B, or 72B versions).
The paper would be much stronger if it could demonstrate robustness across different backbones or scales.

**Questions:**

1. Can you verify whether TimeSearch-R trained with Qwen2.5-VL-7B can work directly with other model variants (e.g., Qwen2.5-VL-3B, 34B, or 72B) without retraining?
If not, how much fine-tuning or adaptation is required?

2. Have you tested whether the learned search policy can generalize to other Video-LLMs beyond Qwen2.5-VL, such as LLaVA-Video or Gemini-style multimodal backbones?

---

> ### Author Response · Authors · 2025-11-27
> **Response to Reviewer mLMd**
>
> The authors thank the reviewer for the valuable time and effort, and provide point-by-point responses as follows:
> ### **1. Policy Transfer and Model Generalization**
> We appreciate the reviewer’s suggestion and have added experiments on policy transfer in **Table 2**. As shown, the search strategy learned by TimeSearch-R transfers effectively to a variety of models, including the proprietary API model GPT-4o and the open-source LLaVA-OV-72B. By using the small RL-trained model to perform temporal search, and then feeding the selected video frames into a larger model to generate the final answer, we significantly reduce inference cost and API invocations while achieving better performance than the large model alone—without requiring any retraining. Under a 32-frame budget, transferring TimeSearch-R’s search policy to GPT-4o and LLaVA-OV-72B yields 2.3% and 3.1% performance improvements on VideoMME. Moreover, compared with the previous state-of-the-art temporal search method T*, TimeSearch-R achieves 9% and 1.2% further improvements under the same transferred settings. These results highlight the strong generalization capability of the learned search strategy across different models. In addition, **Table 3** reports performance on the video-reasoning benchmark Video-Holmes, demonstrating that TimeSearch-R generalizes not only across models but also across tasks.
>
> | Model | # Frame | VideoMME (w/o sub) | MLVU | LongVideoBench |
> |-|:-:|:-:|:-:|:-:|
> | GPT-4o |    8    |        63.9        | 58.1 |      56.7      |
> | LLaVA-OV-72B|    8    |        60.5        | 62.8 |      56.8      |
> | GPT-4o |    32   |        70.8        | 66.6 |      61.1      |
> | LLaVA-OV-72B |    32   |        66.4        | 69.2 |      62.4      |
> | T* + GPT-4o |    8    |        56.5        |   -  |        -       |
> | T* + LLaVA-OV-72B |    8    |        59.0        |   -  |        -       |
> | T* + GPT-4o |    32   |        64.1        |   -  |        -       |
> | T* + LLaVA-OV-72B |    32   |        68.3        |   -  |        -       |
> | TimeSearch-R-7B |   7.8   |        59.1        | 68.6 |      57.1      |
> | TimeSearch-R + GPT-4o |   7.8   |        68.5        | 68.6 |      60.8      |
> | TimeSearch-R + LLaVA-OV-72B |   7.8   |        64.2        | 71.5 |      61.1      |
> | TimeSearch-R-7B |   31.9  |        64.1        | 71.3 |      61.6      |
> | TimeSearch-R + GPT-4o |   31.9  |        73.1        | 69.2 |      63.4      |
> | TimeSearch-R + LLaVA-OV-72B |   31.9  |        69.5        | 72.7 |      64.5      |
>
> ### **2. Training Cost Analysis**
> The end-to-end training process is highly efficient. We use only ~100 samples for SFT and 2.4K samples for RL, and the entire training completes in 10 hours on 32×A100 GPUs, illustrating the practicality and low cost of the proposed method.
>
> We hope the above responses adequately address the reviewer’s concerns. If the reviewers have any other questions, we would be very glad to have further discussions.

---

> ### Comment · Reviewer_mLMd · 2025-11-28
> **final decision**
>
> Thank you for the additional experiments addressing generalization. My concerns have been resolved. I maintain my original score.

---

### Official Review · Reviewer_aDf9 · 2025-11-01

**Soundness:** 3
**Presentation:** 3
**Contribution:** 3
**Rating:** 8
**Confidence:** 4

**Summary:**

The goal is to train models to be able to reason about long-form video, which means an iterative process of search/retrieval paired with state-tracking for eventual answer generation. The authors provide a formulation for training first via SFT and also an updated RL formulation.

**Strengths:**

Long-form video understanding and reasoning is an important task on which models perform poorly.  This approach aims to move the needle and leverages a nice balance of search and "reasoning" (as CoT is usually done) to integrate the video content where appropriate to validate a hypothesis (the question). The approach appears to work well and is intuitive.

**Weaknesses:**

1. See questions below
2. I fear that I'm missing some intuition about where to go next after this paper. I'm trying to use the examples in the appendix (Fig 16/17) to reason about where/why things fail and if there's anything that can be done about them.  The author's guidance on the importance of specific parameters (e.g. 8) would be appreciated but also where the approach is not appropriate.  For example, in the insufficient search, is the data well characterized such that we know how well an oracle would do and how often the number of data points is out of scope for the budget, the model, etc? Do some tasks reduce to exhaustive search? Are there other aspects of visual reasoning/continuity/etc that are missing from models such that even with the perfect set of frames the model cannot succeed?  I would like to be able to factor the various components/concerns and understand their interplay.

minor
- L128, prue

**Questions:**

- Can you enumerate some degenerate solutions the training is likely to find? And what mitigation was required (or if none, why didn't we end up with reward hacking)? My first thought was something about exhaustive search, wide temporal windows, etc
- I don't understand the role of 0.5 in (4),
- Fig 3 makes it seem like there are a set of answers one per frame in the retrieval? But many intermediate clips would not align with the actual final answer, correct?
- Can a distribution of the range of frames (8.8 being average) be provided? More generally, insight into which frames are selected, why, and/or what biases the count?
- L323 says there's a budget of 8, but table says 8.8 average? Also, how was this value determined?
- Can Table 2 be easily run with smaller budgets for a more direct comparison to prior work?

---

> ### Author Response · Authors · 2025-11-27
> **Response to Reviewer aDf9**
>
> The authors appreciate the reviewer’s rigorous assessment and constructive comments, and provide point-by-point responses as follows:
> ### **1. Degenerate Solutions and Mitigation Strategies**
> We indeed observed several types of degenerate behaviors during training and applied effective mitigation strategies as follows:
> 1. **Exhaustive Search**: This behavior appeared more frequently in early training, where the model repeatedly searched without providing an answer. Because the completeness reward constrains both the correctness of the final answer and intermediate search steps, the model quickly learned to perform only the necessary search. The table below summarizes the proportion of exhaustive-search cases before and after RL across different benchmarks, showing a clear reduction and validating the effectiveness of the CSV reward.
> 2. **Wide Temporal Window**: This phenomenon occurred more often on short videos. To mitigate it, we selected longer videos for training (**Appendix Figure 7** shows an average training video length of 1659s), and filtered out questions solvable with only four frames. Combined with the system prompt constraint `MAX_NUM_FRAMES_PER_TURN = 8` (see **Appendix C**) and the CSV design, the model was compelled to locate truly informative segments in long videos, thereby avoiding overly broad temporal windows.
> |exhaustive search rate|VideoMME|MLVU|LongVideoBench|Video-Holmes|
> |-|:-:|:-:|:-:|:-:|
> |before RL|3.85%|6.71%|15.77%|8.49%|
> |after RL|0.11%|0.37%|0.97%|0.16%|
> ### **2. Correctness Threshold of 0.5**
> We follow the setting of Video-R1, using 0.5 as the threshold for determining correctness in video QA. Some benchmarks use open-ended questions, where accuracy can be a continuous value in [0, 1]. In such cases, 0.5 provides a reasonable cutoff for correctness. For most benchmarks, however, questions have definitive ground-truth answers (e.g., multiple choice), meaning the true accuracy is binary, and the 0.5 threshold is directly applicable.
> ### **3. Alignment Between Intermediate Clips and Final Answers**
> **Figure 3** illustrates the workflow of GRPO and CSV. In GRPO, each rollout samples a set of parallel outputs. Under interleaved text–video reasoning, each output consists of textual reasoning, selected video clips, and a final answer. These outputs differ from one another—some sampled video segments may not align with their corresponding final answers. The final reward is computed based on the relative advantage across all sampled outputs, not on a single trajectory.
> ### **4. Distribution of Searched Frame**
> Following the reviewer’s suggestion, we added the distributions of search turns and retrieved frames in **Appendix Figures 14 and 15**. The results show that most questions require only one or two search rounds. Longer videos and more complex reasoning tasks require more frames, indicating that the model has learned an adaptive search strategy conditioned on both the data and task difficulty.
> ### **5. Aligned Average Frame Count**
> We appreciate the reviewer for pointing out this issue. In the revised **Table 1**, we enforce the correct average-frame constraint and report TimeSearch-R’s performance with an average of 7.8 retrieved frames. Despite this lower frame budget, TimeSearch-R still achieves a temporal F1 of 8.4, more than 3× that of the previous best method T*‘s 2.5 (for a fixed method, fewer keyframes usually imply higher precision and lower recall). The average frame count is derived from the total retrieved frames across all samples divided by the number of samples and can be controlled via parameter setting of system prompt in TimeSearch-R.
> ### **6. Comparisons Under Smaller Budgets**
> In the original submission, we followed Qwen2.5-VL’s optimal frame-budget setting (768) to ensure fairness in comparing with the base model. Following the reviewer’s recommendation, we added experiments with smaller budgets in the revised **Table 2** to more directly compare with prior work. TimeSearch-R remains consistently superior across all benchmarks even under these tighter constraints.
>
> We hope the above responses adequately address the reviewer’s concerns. If the reviewers have any other questions, we would be very glad to have further discussions.

---

### Official Review · Reviewer_Ds7H · 2025-11-01

**Soundness:** 3
**Presentation:** 3
**Contribution:** 2
**Rating:** 4
**Confidence:** 4

**Summary:**

The goal of this paper is to propose an RL framework to solve the temporal search problem in video-language data in a general way without human annotations.
* Solid contribution with practical value: The paper presents a well-executed approach that achieves consistent improvements across downstream Video QA tasks on appropriate benchmarks (VideoMME, MLVU, LVB), demonstrating that the method works broadly. The framework successfully addresses the temporal search problem without requiring human annotations, and the planned release of code, models, and data will facilitate future research in this actively growing area.
* Well-grounded work with clear limitations: While the performance gains are moderate rather than substantial, and the novelty is somewhat incremental (applying GRPO to frame search), the paper is well-motivated, clearly written, technically sound, and appropriately positioned in the literature. The work answers a relevant question ("Does RL-based frame search work without human annotations?") affirmatively with proper empirical support.

**Strengths:**

* The paper is well written and is easy to understand.
* The paper proposes a novel algorithm for frame search that does not rely on human annotations and leverages “thinking” with video frames and text.
* The authors have a fairly good set of eval benchmarks: VideoMME MLVU and LVB.  To better understand when the authors’ work should be used, it would be interesting to see the cases where the base model performs better, or any potential failure modes of the proposed method.
* The code, models, and data being released will help facilitate research in this area, which is actively growing area of research in long video understanding.

**Weaknesses:**

* Performance is not substantially better than existing methods, often only moderate performance gains at best.  However, there are consistent gains on the downstream Video QA task, which indicates the approach can work broadly
* This approach requires finetuning the base VLM via RL, whereas other approaches (e.g., T*) do not require any finetuning and can thus leverage API endpoints.
* The novelty of the method may be somewhat limited in the sense that it applies a known technique, GRPO, to the setting of frame search in videos.

**Questions:**

1. More comprehensive analysis:
Cases where the base model performs better than the proposed method
Clear failure mode analysis
This would demonstrate deeper understanding and make the work more valuable for practitioners deciding when to use this approach
2. Stronger empirical results in at least one of these ways:
More substantial performance gains (not just moderate/consistent)
Demonstrating the method works well on additional benchmarks
Showing the approach scales better or generalizes to new domains
3. Addressing the finetuning limitation:
Either: showing the finetuning requirement is justified by significantly better performance
Or: demonstrating other practical advantages that offset not being able to use API endpoints (e.g., efficiency, cost, control)

---

> ### Author Response · Authors · 2025-11-27
> **Response to Reviewer Ds7H**
>
> The authors appreciate the reviewer’s careful evaluation and valuable suggestions, and provide point-by-point responses as follows:
> ### **1. Comprehensive Case Studies**
> Following the reviewer’s recommendation, we have added a broader set of failure cases in **Appendix Figures 19–24**, including scenarios where the base model outperforms the search model. We also provide an in-depth discussion in **Section 3.4** of the main paper. We categorize the observed failure modes as follows:
> 1. Insufficient exploration caused by repeated reasoning that exhausts the frame budget before answering.
> 2. Temporal misalignment relative to the base model, potentially due to limited size and quality of training data.
> 3. Reasoning inconsistency still exists, though significantly reduced after the E2E RL training.
> 4. Incorrect predictions of frame number leads to irrelevant observations, often attributable to model capacity or reasoning capability.
>
> These failure cases reveal several current limitations and highlight promising future directions, including robust timestamp prediction, efficient reasoning mechanisms, automatic evaluation of reasoning quality, and adaptive frame number prediction.
> ### **2. Strong Empirical Results**
> We address the reviewer’s concern from the perspectives of both performance improvement and method generalization.
>
> **Performance Improvement**
>
> We aligned the experimental setup in **Tables 1 and 2** with prior work. On temporal search tasks, TimeSearch-R achieves a temporal F1 of 8.4, substantially outperforming uniform sampling (2.2) and the prior best temporal-search method T* (2.5). On video understanding tasks, TimeSearch-R consistently and significantly outperforms the base model Qwen2.5-VL, the reasoning model Video-R1, and all existing temporal search methods. With a 32-frame budget, it achieves 64.1% average accuracy on VideoMME, surpassing Qwen2.5-VL and Video-R1 by 2.9% and 4.2%, respectively.
> It also outperforms VideoAgent by 8.1%, which uses far more frames, and matches the performance of T*, which uses the proprietary GPT-4o API. To further understand the sources of improvement, we conducted diagnostic comparisons in **Appendix Tables 6 and 7**. Results indicate that our method yields clear gains in local perception and temporal reasoning, especially on tasks requiring fine-grained temporal analysis and multi-step inference. These improvements stem from the learned dynamic search policy, which captures precise spatiotemporal cues essential for local and temporal reasoning.
>
> **Method Generalization**
>
> As an additional benchmark, **Table 3** reports results on Video-Holmes. With a 32-frame budget, TimeSearch-R outperforms Qwen2.5-VL by *14.4%* and Video-R1 by *5.7%*. When increasing the budget to 768, it even surpasses the advanced proprietary reasoning model Gemini-2.0-Flash-Thinking, demonstrating that adaptive temporal search effectively enhances reasoning and generalizes to new tasks. **Table 2** further shows that the learned search policy transfers naturally to other foundation models. Under a 32-frame budget, applying TimeSearch-R to GPT-4o and LLaVA-OV-72B yields improvements of 2.3% and 3.1% on VideoMME, respectively, and outperforms T* by 9% and 1.2%. These results confirm strong cross-model generalization of the learned policy.
> ### **3. E2E Optimization vs. API Calls**
> Long-video understanding requires multi-frame inputs and iterative search, making inference costly. In contrast to expensive proprietary APIs, TimeSearch-R uses only a 7B model, resulting in substantially lower per-query cost. As shown in **Table 2**, the learned search policy can be transferred to API models: the small model performs the temporal search, while the API model provides the final answer. This hybrid scheme significantly reduces inference cost while improving accuracy. The e2e optimization confers adaptability to downstream tasks—an advantage not achievable through API calls or prompt engineering.
> ### **4. Novelty of the Proposed Method**
> We respectfully clarify that our contribution is not a direct application of GRPO to video search. Instead, we introduce a novel GRPO-CSV algorithm designed specifically to address insufficient temporal exploration and inconsistent logic reasoning caused by unsupervised reasoning steps. CSV introduces intermediate completeness rewards without requiring human annotations. **Section 2.2** of the revised paper includes a more detailed theoretical explanation:
> * From an information-theoretic perspective, CSV increases the mutual information between video inputs and answers, encouraging more complete utilization of visual evidence.
> * From an optimization perspective, CSV transforms sparse correctness rewards into denser signals, enabling better credit assignment and more effective policy optimization.
>
> We hope the above responses adequately address the reviewer’s concerns. If the reviewers have any other questions, we would be very glad to have further discussions.

---

### Author Response · Authors · 2025-11-26
**Response to All Reviewers**

We sincerely thank all reviewers for their time and effort. We are encouraged by the recognition that our work is technically sound (**Reviewer Ds7H**), important and intuitive (**Reviewer aDf9**), and clearly motivated with remarkable results (**Reviewers mLMd, VVki**).

Following the reviewers’ suggestions, we have revised the paper and added more comprehensive experiments and analyses, which are highlighted in blue in the revised PDF:
1. **Expanded Evaluation Benchmarks**: To address concerns from Reviewers Ds7H, mLMd, and VVki regarding model performance and generalization, we evaluated TimeSearch-R on the complex video reasoning benchmark Video-Holmes (**Table 3**). We further provide detailed analyses of the performance gains introduced by TimeSearch-R on VideoEvalPro (**Appendix Table 6**) and VideoMME (**Appendix Table 7**). Additionally, **Table 2** now includes experiments that transfer the learned TimeSearch-R policy to other foundation models.
2. **Aligned Experimental Settings**: In response to Reviewer aDf9’s and VVki's concern about alignment of frame budget with prior work, we refined the number of keyframes in **Table 1** and added comparisons under smaller budgets in **Table 2** to align with previous work.
3. **Statistical Search Behavior**: To address the curiosity of Reviewers aDf9 and VVki regarding the model’s actual search behavior, we included distributions of search turns and frames in different tasks in **Appendix Figures 14-15**. These analyses help reveal potential biases learned during end-to-end RL training and the advantages the model exhibits in different scenarios.
4. **Additional Case Studies**: To meet the request from Reviewers Ds7H, aDf9, and VVki for more comprehensive analyses, we included additional failure-case studies in **Appendix Figures 19–24**. These also cover scenarios where the base model outperforms the temporal search model. We provide an in-depth discussion of limitations and potential future directions.
5. **Detailed Efficiency Analysis**: To address Reviewer VVki’s concern regarding efficiency, we added a thorough analysis of training (CSV phase) and inference efficiency in **Appendix Section E**.
6. **Theoretical Analysis**: In response to Reviewer VVki’s request for more theoretical justification, we expanded the explanation of the **completeness reward** in **Section 2.2**, providing analyses from both information-theoretic and optimization perspectives.
7. **Minor Typos**: We thank Reviewer aDf9 for pointing these out.

We also provided point-by-point responses to each reviewer's questions individually below.

---

### Meta-Review · Area_Chair_4TkA · 2026-01-10

**Summary:**

This paper presents an RL-based temporal search framework for long-form video understanding. The reviews are polarized, featuring two scores of 8 and two scores of 4. This makes it a clear borderline case. **Reviewer Ds7H** and **Reviewer VVki** find the novelty limited because the method adapts existing RL tools to a search setting using heuristics. They question if the added complexity is worth it, especially given the need to fine-tune a specific backbone.

By contrast, **Reviewer aDf9** and **Reviewer mLMd** find the approach practical and note consistent gains across multiple benchmarks. These reviewers appreciate the focus on long-form video reasoning and the clear problem setup. During the rebuttal, the authors added failure mode analyses, temporal F1 comparisons, and cross-model generalization experiments. These additions address many concerns regarding robustness and practical utility.

While the conceptual novelty remains moderate, the empirical value is clear. The paper offers a well-documented approach for an important application area. I lean toward a weak-accept recommendation for an ICLR poster.

**Reviewer Concerns:**

**Reviewer Ds7H** initially worried about modest gains and the complexity of RL fine-tuning. They also requested better failure mode analysis and evidence of generalization. In response, the authors added a temporal F1 comparison against a T* baseline and tested their policy on GPT-4o and LLaVA-OV-72B. They also included results from the Video-Holmes benchmark. These additions show the method is more robust than first thought.
The authors also justified the cost advantages of using an open-source 7B model instead of proprietary APIs. This helps with the practicality concern, but the core issue of conceptual novelty remains. The method is a careful adaptation of existing RL techniques rather than a fundamentally new algorithm. I believe the new results make the empirical case stronger, even if the underlying contribution is still an application of known ideas.

**Reviewer aDf9** liked the integration of search and reasoning but asked for more insight into design choices. The authors provided details on the reward design to show how it prevents degenerate solutions. They also explained the role of specific constants and provided data on frame distributions. These clarifications strengthened the paper and addressed the technical questions about search budgets.
Most of this reviewer's issues involved missing analysis rather than fundamental flaws. The rebuttal addressed these points directly with added experiments. I expect **Reviewer aDf9** to remain positive given these improvements.

**Reviewer mLMd** focused on whether the search policy was overfit to the Qwen2.5-VL-7B backbone. The rebuttal countered this with cross-model generalization experiments. These tests showed the policy can improve performance for GPT-4o and LLaVA-OV-72B. This evidence suggests the approach is not limited to a single model architecture.
The reviewer's main concern was model coupling and the authors provided the exact evidence requested. A broader study across more architectures would be better, but the current evidence is convincing for ICLR standards. I expect the score to stay at 8.

**Reviewer VVki** remained critical of the heuristic nature of the completeness reward and the lack of theoretical proofs. The authors added empirical ablations and clarified their design, but they did not provide a formal convergence analysis. While the practical results are strong, this reviewer’s concerns about theoretical grounding are still valid.
The rebuttal improved the empirical understanding of the method but did not change its conceptual nature. I do not expect **Reviewer VVki** to change their overall stance. The method remains an incremental extension of existing frameworks in their view.

**Reviewer Scores:**

- **Reviewer Ds7H (Original: 4 → Predicted: 6)**
  New failure mode analysis and temporal F1 comparisons speak directly to the original concerns. The cross-model results show the gains are more robust than initially suggested. I expect an upward revision to a weak-accept level.

- **Reviewer aDf9 (Original: 8 → Predicted: 8)**
  The rebuttal addressed technical and analytical questions with new experiments on reward design and search budgets. The score should remain at an accept level.

- **Reviewer mLMd (Original: 8 → Predicted: 8)**
  Transfer experiments to GPT-4o and LLaVA-OV-72B addressed the main concern about model coupling. The score should stay at 8 with higher confidence.

- **Reviewer VVki (Original: 4 → Predicted: 4)**
  This reviewer looks for theoretical guarantees and a more principled framework. The rebuttal provided empirical stability results but no new theory, so the score will likely stay at 4.

The post-rebuttal picture shows a clear leaning toward acceptance with three positive scores and one negative.

---

### Decision · Program_Chairs · 2026-01-26

Accept (Poster)